# `CE-NAS`: An End-to-End Carbon-Efficient Neural Architecture Search Framework

**Yiyang Zhao**
Worcester Polytechnic Institute

**Yunzhuo Liu**
Shanghai Jiao Tong University

**Bo Jiang**
Shanghai Jiao Tong University

**Tian Guo**
Worcester Polytechnic Institute

## Abstract

This work presents a novel approach to neural architecture search (NAS) that aims to increase carbon efficiency for the model design process. The proposed framework `CE-NAS` addresses the key challenge of high carbon cost associated with NAS by exploring the carbon emission variations of energy and energy differences of different NAS algorithms. At the high level, `CE-NAS` leverages a reinforcement-learning agent to dynamically adjust GPU resources based on carbon intensity, predicted by a time-series transformer, to balance energy-efficient sampling and energy-intensive evaluation tasks. Furthermore, `CE-NAS` leverages a recently proposed multi-objective optimizer to effectively reduce the NAS search space. We demonstrate the efficacy of `CE-NAS` in lowering carbon emissions while achieving SOTA results for both NAS benchmarks and open-domain NAS tasks. For example, on the HW-NasBench, `CE-NAS` reduces carbon emissions by up to 7.22X while maintaining a search efficiency comparable to vanilla NAS. For open-domain NAS tasks, `CE-NAS` achieves SOTA results with 97.35% top-1 accuracy on CIFAR-10 with only 1.68M parameters and a carbon consumption of 38.53 lbs of $CO_2$. On ImageNet, our searched model achieves 80.6% top-1 accuracy with a 0.78 ms TensorRT latency using FP16 on NVIDIA V100, consuming only 909.86 lbs of $CO_2$, making it comparable to other one-shot-based NAS baselines. Our code is available at https://github.com/cake-lab/CE-NAS.

## 1 Introduction

Deep Learning (DL) has become an increasingly important field in computer science, with wide applications such as healthcare and finance [81, 6, 60, 30, 33, 63, 35, 38]. Neural architecture search (NAS) has emerged as a means to automate the design of DL models, which involves training *many DL models* from a massive architecture design space with hundreds of millions to trillions of candidates [102, 66, 79, 97, 98, 46]. Consequently, NAS can be energy-intensive and significantly contributes to today's carbon emissions [68, 102]. Many NAS works have reported using thousands of GPU-hours [102, 66, 79, 78, 98]. For example, Zoph et al. [102] used 800 GPUs for 28 days, resulting in 22,400 GPU-hours, to obtain the final architectures. Strubell et al. [68] found that a single NAS solution can emit as much carbon as five cars during its lifetime.

The environmental impact of NAS, if left untamed, can be substantial. While recent works have significantly improved the search efficiency of NAS [102, 66, 79, 78, 97], e.g., reducing the GPU-hours to tens of hours without sacrificing the architecture quality, there still lacks *conscious efforts in reducing carbon emissions*. As noted in a recent vision paper by Bashir et al. [9], energy efficiency can help reduce carbon emissions but is not equivalent to carbon efficiency. This paper aims to

38th Conference on Neural Information Processing Systems (NeurIPS 2024).

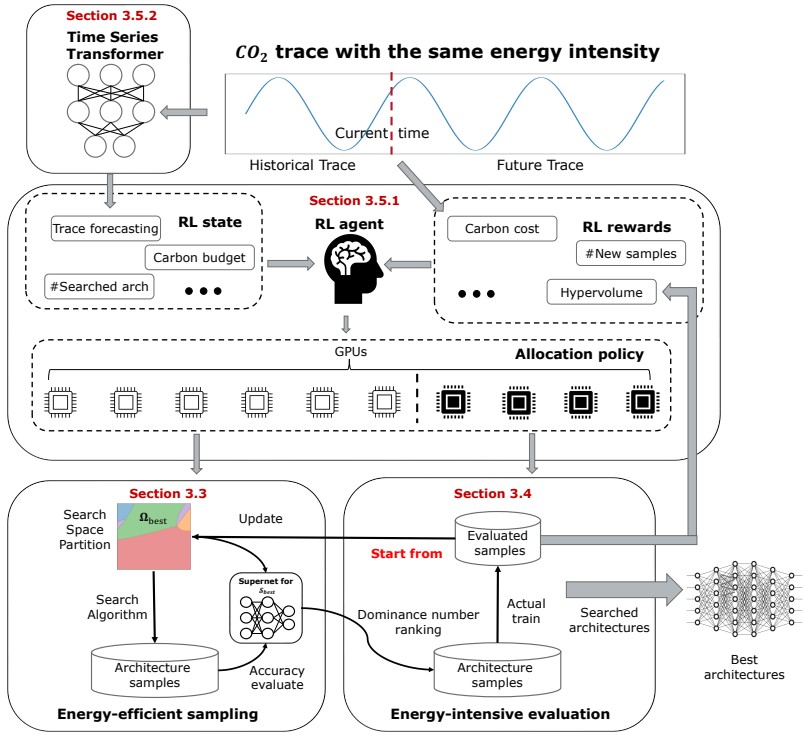

Figure 1: An overview of CE-NAS. The sampling and evaluation tasks will be allocated different GPU resources based on carbon emission intensity during the neural architecture search. Specifically, when the $CO_2$ intensity is low, we allocate more resources to energy-intensive evaluation, which focuses on assessing the performance of sampled architectures generated through energy-efficient sampling. Conversely, when $CO_2$ intensity is high, more GPUs are dedicated to energy-efficient sampling, which utilizes various NAS algorithms to sample new architectures from a search space partitioned based on previously evaluated architectures. The specific GPU allocation strategy is learned through a reinforcement learning agent, as detailed in the middle section of the figure.

bridge the gap between carbon and energy efficiency with a new NAS framework designed to be carbon-aware from the outset.

The proposed framework, termed CE-NAS, tackles the high carbon emission problem from two main aspects. First, CE-NAS regulates the model design process by *deciding when to use different NAS evaluation strategies based on the carbon intensity*, which varies geographically and temporally with the mix of active generators as observed in [9]. To elaborate, given a fixed amount of GPU resources, CE-NAS will allocate more resources to energy-efficient NAS evaluation strategies, e.g., one-shot NAS [65, 10, 46, 97, 14, 13], during periods of high carbon intensity. Conversely, during periods of low carbon intensity, CE-NAS will shift the focus to running energy-intensive but more effective NAS evaluation strategies, e.g., vanilla NAS [102, 66, 79, 78]. We design and train a reinforcement learning (RL) agent (§3.5) based on historical and predicted carbon emissions (§3.5.2) and observed search results to make such allocation decisions. Second, the CE-NAS framework will support searching for DL models satisfying multiple metrics beyond the commonly used metric of inference accuracy. CE-NAS can consider more metrics such as inference latency, #PARAMs, #FLOPs, etc. For example, CE-NAS can search for models with high accuracy and low inference latency. Specifically, both the search and deployment efficiency are achieved by integrating a recent learning-based multi-objective optimizer LaMOO [98] to CE-NAS. Figure 1 depicts the overall workflow of CE-NAS.

To demonstrate CE-NAS's efficacy in addressing the energy and carbon issues, we conduct a comprehensive evaluation using both commonly used NAS benchmarks and real-world NAS tasks. For example, utilizing carbon emission data from ElectricityMap [58], CE-NAS achieves better performance with reduced carbon costs compared to existing methods on various NAS benchmarks, including HW-NASBench [40] and NasBench301 [93]. In open-domain tasks, CE-NAS achieves the state-of-the-art (SOTA) top-1 accuracy of 97.35% with only 1.68M parameters on the CIFAR-10

image classification task, with $CO_2$ costs similar to those of one-shot-based NAS algorithms. For the ImageNet dataset, `CE-NAS` achieves the SOTA top-1 accuracy of 80.6% under the same inference latency of 0.78 ms with TensorRT on NVIDIA V100, while maintaining a carbon cost comparable to SOTA NAS methods [13]. We also evaluate our carbon forecasting model and show that it outperforms existing models [11, 56, 55], including the SOTA method [55].

In summary, we make the following key contributions.

- We introduce a carbon-efficient NAS framework named `CE-NAS` that can dynamically allocate GPU resources among different NAS evaluation methods—namely, vanilla NAS and one/few-shot NAS—based on carbon emissions data and current search results. `CE-NAS` is an end-to-end framework that synergistically integrates an RL-based agent for GPU resource allocation, a time-series transformer [2] for carbon intensity prediction, and a multi-objective optimizer [98] for reduced search space.

- We implement and evaluate `CE-NAS` on different NAS benchmarks, including HW-NasBench [40] and Nasbench301 [93]. We show that `CE-NAS` can achieve the best search performance compared to only using vanilla NAS, one-shot NAS, and a recently proposed heuristic GPU allocation strategy [4] given the same carbon budget.

- `CE-NAS` delivers SOTA architectural performance in open-domain NAS tasks while maintaining carbon costs comparable to one-shot NAS methods. For example, on CIFAR-10, `CE-NAS` achieves a top-1 accuracy of 97.35% with only 1.68M parameters and a carbon cost of 38.53 lbs $CO_2$. On ImageNet, `CE-NAS` reaches a top-1 accuracy of 80.6% with a 0.78 ms TensorRT latency using FP16 on NVIDIA V100.

## 2 Related Work

***Efficient neural architecture search (NAS)*** often focuses on improving the evaluation phase, e.g., via weight-sharing (one-shot) [65, 46, 88, 18, 13], zero-shot proxy [41, 85, 3, 37, 73, 76, 95, 74, 50, 59, 44, 70, 16, 17], performance predictor [45, 13, 79, 93], low-fidelity NAS evaluation [102, 66, 79, 78, 46, 36], and gradient proxy NAS [87]. A comparison of these methods can be found in Table 4 in Appendix. Weight-sharing leverages the accuracy estimated by a *supernet* as a proxy for the true architecture performance, while gradient proxy NAS uses the gradient as a proxy. These proxy-based methods, although incurring smaller search costs in terms of energy, can have lower search efficiency because their estimated architecture accuracy may have poor rank correlation [97]. Zero-shot proxy NAS eliminates the need for *supernet* training entirely, serving as a fully training-free proxy for network evaluation. However, as noted in [41], mainstream zero-shot NAS methods still struggle to achieve high rank correlations between predicted and true accuracies, even when compared to weight-sharing-based NAS evaluation. Performance predictors provide a more accurate performance prediction than weight-sharing and gradient proxy NAS. Still, their accuracy heavily relies on the volume and quality of the training data, which can be very expensive to create [90, 93]. Low-fidelity evaluation still requires training each searched architecture, leading to limited energy savings. In practice, simply employing these methods without modification might not effectively balance carbon emissions and performance during the search process. `CE-NAS` successfully achieves good search efficiency, search quality, and carbon efficiency by integrating the weight-sharing NAS method into vanilla NAS. We also leverage the low-fidelity evaluation method in the open-domain NAS search.

***The high carbon emission of NAS*** is a pressing issue. A study by [68] found that a single NAS solution can emit as much carbon as five cars over its lifetime. For example, if we design a transformer-based model, training a base Transformer model can use 96 GPU hours on NVIDIA's P100 [68], while training a larger model can take 28 GPU days. A recent NAS work [68] on transformer revealed that their comprehensive architecture search used 979 million training steps, totaling 274,120 GPU hours and resulting in 626,155 pounds of $CO_2$ emissions. Although several NAS studies [87, 13, 12] have attempted to address carbon issues during the search process, they often treat carbon emissions uniformly, regardless of time and location variances. These carbon-agnostic NAS methods therefore miss the opportunity to explore the inherent carbon variations [9]. Concurrently, systems researchers have explored the temporal and spatial variations exhibited in electricity's carbon emission for popular data center applications like training [4, 9] and inference [39]. This work targets a special form of training workload, i.e., NAS, by leveraging the domain knowledge to optimize the end-to-end NAS process to explore carbon variations. Specifically, we concentrate on implementing the temporal

variations in carbon intensity rather than spatial variations, as managing the carbon costs associated with spatial shifts poses significant challenges. Transferring large volumes of data over the Internet can incur substantial carbon costs, and accurately estimating these costs is complex. Data often passes through numerous intermediate routers across large geographic regions, making it difficult to ensure that the benefits of a spatial shift outweigh its associated overhead.

# 3 The Design of a Carbon-Efficient NAS Framework

In this section, we present an overview of the proposed `CE-NAS` framework (§3.1). We first introduce the initial setup of our `CE-NAS` in §3.2. We then propose a sampling strategy based on the one-shot/few-shot NAS evaluation method in §3.3. Finally, we discuss the architecture evaluation part in §3.4, which is both time and computing resource-intensive. In §3.5, we describe how to use an RL-based agent to automatically allocate GPU resources based on predicted hourly carbon intensity.

## 3.1 `CE-NAS` Overview

As observed in [9], grid carbon emissions vary geographically and temporally based on the mix of active generators. Consequently, systems can emit different amounts of carbon even when consuming the same electricity at different locations or times. Currently, neither one (few)-shot nor vanilla NAS methods account for these variations in carbon emissions, which can lead to inefficiencies in terms of NAS carbon usage. Furthermore, one (few)-shot and vanilla NAS exhibit different search and carbon efficiency trade-offs. Vanilla NAS is carbon-intensive but effective, while one (few)-shot NAS can be carbon-friendly but sample-inefficient.

`CE-NAS` is a NAS framework that directly addresses the high-carbon issues by balancing energy consumption across high-carbon and low-carbon periods. The primary objective of this framework is to efficiently search for optimal neural architectures while minimizing the carbon footprint associated with the NAS process. Specifically, `CE-NAS` decouples the two parts of a NAS search process—energy-efficient sampling and energy-intensive evaluation—and handles them independently across different carbon periods. For each carbon period, `CE-NAS` uses an RL-based agent to automatically allocate GPU resources for the NAS search process to achieve good carbon efficiency.

## 3.2 Search Initialization

`CE-NAS` considers more than one search objective(e.g., accuracy, mAP, latency, #FLops, etc), so we formulate the search problem as a multi-objective optimization (MOO). Similar to other NAS and optimization problems [24, 77, 98, 66], `CE-NAS` initializes the search process by randomly selecting several architectures, $\mathbf{a}$, from the search space, $\Omega$, and evaluating their accuracy, $E(\mathbf{a})$, and other deployment metrics (#Params, #FLops, inference latency, inference energy, etc) $T(\mathbf{a})$. Compared to the accuracy $E(\mathbf{a})$, $T(\mathbf{a})$ can be quickly measured without much cost. The resulting samples are then added to the observed samples set, $\mathcal{P}$.

We distinguish two types of methods for evaluating the accuracy of an architecture. One is actual training, which trains an architecture $a$ from scratch until convergence and evaluates it to obtain its true accuracy, $E(a)$. To optimize computing resources during this process, we implement the low-fidelity training strategy described in [102, 66, 46, 79, 36]. This strategy reduces computing costs by simplifying the parameters of neural architectures (e.g., #depth, #channels, #resolutions) or shortening the training process under controlled conditions. The other method is using *one-shot evaluation* [10, 65, 46], which leverages a trained *supernet*, or a zero-shot-based NAS evaluator [95, 74, 50, 59, 44, 70, 16, 17] as a performance proxy to estimate the accuracy of the architecture, denoted as $E'(a)$. Note that obtaining $E'(a)$ is very cheap and carbon-efficient; however, due to the co-adaption among operations [97], $E'(a)$ is often not as accurate as $E(a)$. We train all the sampled architectures in the initialization stage to obtain their true accuracy for further search.

## 3.3 Energy-Efficient Architecture Sampling

**Multi-objective search space partition.** We leverage the recently proposed multi-objective optimizer called LaMOO [98] that learns to partition the search space from observed samples to focus on

promising regions likely to contain the Pareto frontier. LaMOO is an optimizer for general black-box optimization problems; we can apply it to NAS as follows.

We utilize LaMOO [98] to partition the search space $\Omega$ into several subspaces, and find the optimal subspace denoted by $\Omega_{best}$. Next, we construct and train a supernet [10, 97], $\mathcal{S}_{best}$, for $\Omega_{best}$. We then use a NAS search algorithm to identify new architectures that will be used to refine the search space. In this work, we employ the state-of-the-art multi-objective Bayesian optimization algorithm qNEHVI [25]. This algorithm will generate new architectures $a_n$ from $\Omega_{best}$, and estimate their approximate accuracy $E'(a_n)$ using $\mathcal{S}_{best}$. At the same time, these architectures $a_n$ are added to a ready-to-train set $\mathcal{R}$, consisting of candidates for further actual training as described in §3.4.

We define the maximum capacity of $\mathcal{R}$ as $Cap(\mathcal{R})$, hyperparameter in CE-NAS. Note that the architectures in $\mathcal{R}$ are removed once they are actual trained as described in §3.4. When the capacity is reached, i.e., when there are more architectures to train than we have resources for, the sampling process blocks until spaces free up in $\mathcal{R}$. The accuracy of architectures estimated by $\mathcal{S}_{best}$ will be fed back into the search algorithm as shown in Figure 1 to repeat the process described above.

As mentioned in §3.2, obtaining estimated accuracy through supernet is energy-efficient because these architectures can be evaluated without the time-consuming training. Therefore, during high carbon emission periods, CE-NAS will try to perform this process to save energy and produce as little carbon as possible, as shown in the *energy-efficient sampling* part of Fig.1.

### 3.4 Energy-Intensive Architecture Evaluation

If we perform the entire NAS only using the process described in §3.3, CE-NAS will be essentially performing one/few-shot NAS only within the subspace $\mathcal{S}_{best}$. Although these methods are efficient, they typically underperform compared to vanilla NAS. As Zhao et al. showed, it is possible to improve LaMOO's space partition with more observed samples through actual training [98]. This section describes the process to evolve $\mathcal{S}_{best}$ during low carbon emission periods.

At the high level, we will pick new architectures from the ready-to-train set $\mathcal{R}$ to train to convergence and use them to refine the search space partition. That is, the architecture $a$, with its true accuracy, $E(a)$, will be added to the observed sample set $\mathcal{P}$ to help identify a more advantageous subspace, $\Omega_{best}$, for the architecture sampling process as described in [98]. In this work, we sort the architectures in the ready-to-train set $\mathcal{R}$ by their *dominance number*. The dominance number $o(a)$ of an architecture $a$ is defined as the number of samples that dominate $a$ in search space $\Omega$:

$$o(a) := \sum_{a_i \in \Omega} \mathbb{I}[a_i \prec_f a, \, a \neq a_i]^1,  \tag{1}$$

where $\mathbb{I}[\cdot]$ is the indicator function. With the decreasing of the $o(a)$, $a$ would be approaching the Pareto frontier; $o(a) = 0$ when the sample architecture $a$ is located in the Pareto frontier. The use of dominance number allows us to rank an architecture by considering both the estimated accuracy $E'(a)$ and other metrics $T(a)$ at the same time. CE-NAS will first train the architectures with lower dominance number values when GPU resources are available. Once an architecture is trained, it is removed from $\mathcal{R}$.

This process is depicted in the *energy-consuming evaluation* component of Figure 1. Note that this process includes actual time-consuming DL model training, which is time-consuming and highly energy-intensive, leading to significant $CO_2$ emissions. Hence, CE-NAS will try to prioritize this process during periods of low carbon intensity.

### 3.5 Carbon-Efficient GPU Allocation

The carbon impact of the above two processes (i.e., §3.3 and §3.4) in a NAS search is materialized through the use of GPU resources. A key decision CE-NAS needs to make is *how* to allocate GPUs among these two interdependent processes. Assigning too many GPUs to architecture sampling could impact the search efficiency, i.e., the searched architectures are far from the true Pareto frontier; assigning too many GPUs to architecture evaluation could significantly increase energy consumption and carbon emission. CE-NAS must consider these trade-offs under varying carbon intensity and

---

[1]We define *dominance* $y \prec_f x$ as $f_i(x) \leq f_i(y)$ for all functions $f_i$, and exists at least one $i$ s.t. $f_i(x) < f_i(y)$, $1 \leq i \leq M$, where $M$ is the number of objectives.

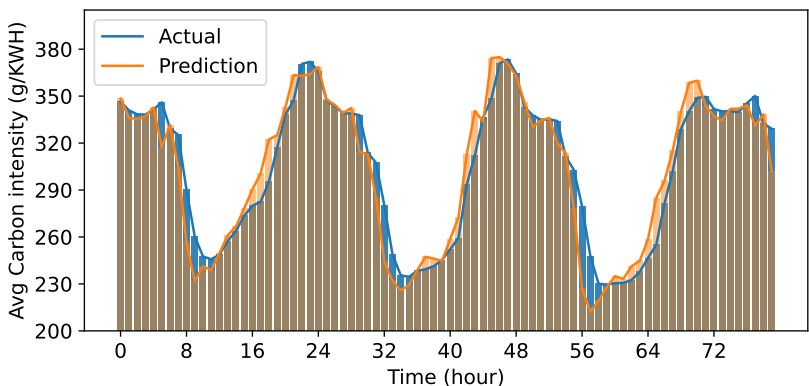

Figure 2: An overview of `CE-NAS`. This carbon trace is based on the US-CAL-CISO data from 2021, specifically covering the period from 0:00, July 2, 2021, to 8:00, July 4, 2021. The blue trace is its actual carbon trace and the yellow trace is the prediction trace by our carbon predictor described in sec. 3.5.2

re-evaluate the GPU allocation strategy. We design an RL-based strategy (§3.5.1) to automatically allocate GPU resources based on predicted carbon intensity in every one hour (§3.5.2).

### 3.5.1   Design of RL-Based GPU Allocation Strategy

Although the heuristic-based strategy proposed by [4] can adjust the tendency for sampling and evaluation based on carbon intensity, it is still far from making the most carbon-efficient decision. It overlooks critical influencing factors such as the varying gains in accuracy from performing sampling and evaluation at different stages of the search. These factors are specific to the NAS algorithms and are challenging to model, which complicates its inclusion in heuristic strategy design. To address this issue, we introduce a method based on reinforcement learning that automatically accounts for these diverse factors to develop allocation strategies. We propose a Reinforcement Learning (RL)-based method to automatically customize a GPU allocation strategy for specific NAS algorithms. In this paper, we choose LaMOO [98] with qNEHVI [25].

The key to applying RL lies in the design of effective state inputs, actions, and rewards according to the specific task. Our `CE-NAS` focuses on tailoring these elements for carbon-efficient NAS tasks, with the guiding principle of simplifying the state and action spaces to expedite efficient and rapid training. Specifically, Our RL agent, which is a simple-structured neural network with only four fully connected layers, takes the remaining carbon budget, the number of already searched architectures and their hypervolume (The definition of the hypervolume refers to Appendix E.2.4.), as well as the future carbon traces predicted by the time transformer as input. Then it outputs the probabilities of different GPU allocation ratios for architecture evaluation as action. These possible allocation ratios are discretized to boost learning efficiency. After the action is executed, `CE-NAS` generates a reward based on the improvement of the performance of the already searched architectures and then uses the reward to refine the policies of the agent by updating its network parameters with the *actor-critic* [61] policy gradient algorithm. The specifics of our RL design can be found in Appendix C and the middle part of Fig. 1.

### 3.5.2   Design of Transformer-Based Carbon Intensity Forecasting

We utilize a machine learning model to predict future carbon intensity, which serves as the input of the RL agent. We leverage a time series transformer-based model, as shown in the top left part of Fig. 1, to forecast future carbon intensity because of their efficacy [62, 84, 99]. Specifically, we leverage the architecture design by employing a standard encoder-decoder Transformer for time series forecasting [2]. This method incorporates temporal features that act as positional encodings within the Transformer's encoder/decoder framework. Past values are input into the encoders, and future values are input into the decoders in the training stage. For instance, as described in [2], if a time series data point corresponds to the 11th of August, the temporal feature vector would be (11, 8), where 11 denotes the day of the month, and 8 signifies the month of the year. We leverage the SOTA

Table 1: Daywise MAPE comparison of our time-series transformer and other baseline methods.

| Region | Day-1 Forecast | | | | Day-2 Forecast | | | | Day-3 Forecast | | | |
|---|---|---|---|---|---|---|---|---|---|---|---|---|
| | STCF [11] | DACF [56] | CC [55] | ours | STCF [11] | DACF [56] | CC [55] | ours | STCF [11] | DACF [56] | CC [55] | ours |
| CISO | 10.71 | 6.45 | 8.08 | **5.26** | 18.99 | 12.26 | 11.19 | **8.69** | 25.24 | 16.02 | 12.93 | **10.83** |
| DE | 15.54 | 7.21 | 7.81 | **5.34** | 31.56 | 11.82 | 10.69 | **8.23** | 42.16 | 13.95 | 12.80 | **10.68** |
| PJM | 4.27 | 3.08 | 3.69 | **2.58** | 7.11 | 5.51 | 4.93 | **3.53** | 8.90 | 7.06 | 5.87 | **4.02** |

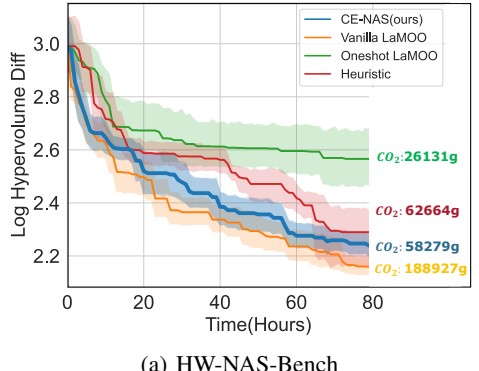

(a) HW-NAS-Bench

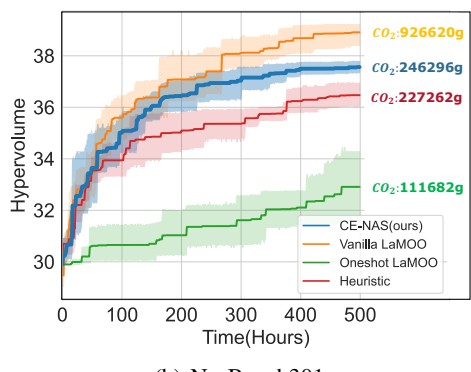

(b) NasBench301

Figure 3: Search progress over time. `CE-NAS` has the second lowest relative carbon emission while achieving the second best $HV_{\text{log\_diff}}$ on HW-NAS-Bench, and `CE-NAS` has the second lowest relative carbon emission while achieving the second best $HV$ on NasBench301.

NAS algorithm LaMOO [98] to design the architecture of the time series transformer. The details of search space and founded architectures are in Appendix D.

## 4 `CE-NAS` Experimental Evaluation

We develop a prototype of the `CE-NAS` framework as outlined in Section 3. This section analyzes `CE-NAS`'s carbon efficiency and search efficacy through trace-driven simulations and emulation. We first demonstrate the performance of our designed time-series transformer for carbon forecasting in § 4.1. We then assess `CE-NAS` across two distinct NAS scenarios: the first leveraging two commonly used NAS benchmarks, HW-NAS-Bench [40] and NasBench301 [93], and the second including real-world computer vision applications, such as image classification.

Our experiments utilize carbon trace data sourced from ElectricityMap [58], an independent carbon information service. We selected the CISO[2] trace beginning on July 2, 2021, due to its variable carbon intensity, which provides an opportunity to evaluate `CE-NAS`'s performance over time and its adaptability to fluctuating carbon levels. Figure 2 illustrates the initial 80 hours of the carbon trace, demonstrating the actual data against the forecasted trace predicted by our designed time series transformer, as detailed in Section 3.5.2. Throughout the `CE-NAS` search phase, predicted carbon intensity is utilized for GPU allocation, while the actual carbon trace informs the calculation of carbon costs in the evaluation phase.

We also conduct several ablation studies detailed in Appendix F. These studies include variations of hyperparameters in the RL agent settings, validation of LaMOO's effectiveness, and a comparison of search performance using predicted versus actual carbon traces, among others. Our findings reveal only a minimal difference in search performance between using predicted and actual carbon traces.

### 4.1 Time-series Transformer for Carbon Forcasting

We assess the performance of our time series forecasting transformer across three distinct regions: CISO, DE[3], and PJM[4]. We compare our approach against other SOTA models referenced in [11, 56,

---
[2]CISO: California Independent System Operator
[3]DE: Germany
[4]PJM: Pennsylvania-Jersey-Maryland Interconnection

Table 2: Search Results on CIFAR-10 using the NasNet search space. The two optimization objectives we are searching for are #params and accuracy.

| Method | Test Error(%) | #Param (M) | Search&Training Cost (GPU Hours)[†] | $CO_2$ (lbs) | NAS Method |
|---|---|---|---|---|---|
| PNAS [45] | 3.41±0.09 | 3.2 | 5400 | 3836.62 | vanilla |
| NAO [54] | 3.14±0.09 | 3.2 | 5400 | 3836.62 | vanilla |
| NASNet-A [102] | 2.65 | 3.3 | 48000 | 30534.78 | vanilla |
| LEMONADE [29] | 2.58 | 13.1 | 2160 | 1514.07 | vanilla |
| AlphaX [79] | 2.54±0.06 | 2.83 | 24000 | 16196.72 | vanilla |
| AmoebaNet-B-small [66] | 2.50±0.05 | 2.8 | 75600 | 43972.18 | vanilla |
| BayeNAS [100] | 2.81±0.04 | 3.4 | 31.2 | 21.70 | one-shot |
| DARTS [46] | 2.76±0.09 | 3.3 | 50.4 | 34.04 | one-shot |
| MergeNAS [80] | 2.68±0.01 | 2.9 | 40.8 | 27.01 | one-shot |
| One-shot REA | 2.68±0.03 | 3.5 | 44.4 | 29.33 | one-shot |
| CNAS [43] | 2.60±0.06 | 3.7 | 33.6 | 23.19 | one-shot |
| PC-DARTS [88] | 2.57±0.07 | 3.6 | 33.6 | 23.19 | one-shot |
| Fair-DARTS [21] | 2.54±0.05 | 3.32 | 98.4 | 65.87 | one-shot |
| P-DARTS [18] | 2.50 | 3.4 | 33.6 | 23.19 | one-shot |
| **CE-Net-P1** | 2.65±0.03 | **1.68** | 228.6 | 38.53 | hybrid |
| **CE-Net-P2** | **2.16±0.06** | 3.30 | 228.6 | 38.53 | hybrid |

[†] The total cost includes both the NAS search duration and the training time for the architectures identified. When applying one-shot NAS to search architectures on CIFAR-10, we estimate that training these architectures from scratch until convergence requires approximately 26.4 GPU hours.

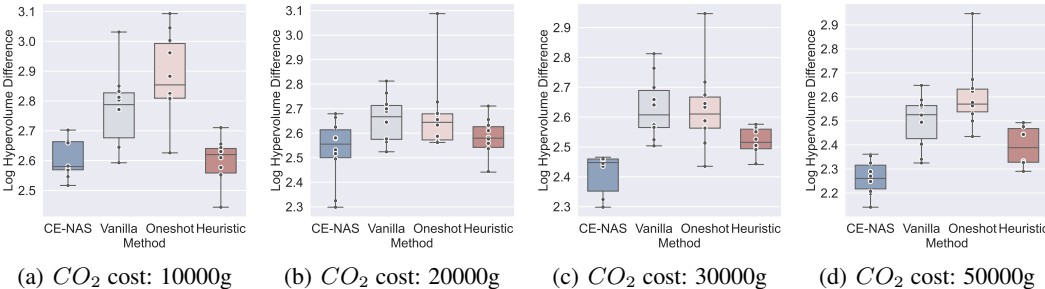

(a) $CO_2$ cost: 10000g    (b) $CO_2$ cost: 20000g    (c) $CO_2$ cost: 30000g    (d) $CO_2$ cost: 50000g

Figure 4: Search efficiency under carbon emission constraints. These results are based on HW-NAS-Bench with carbon trace showing in fig. 2, and we ran each method ten times.

55]. The studies in [11, 56] employ a Feed-Forward Neural Network (FFNN) for their forecasts, while [55] utilizes a combination of FFNN, Convolutional Neural Network, and Long Short-Term Memory networks for multi-day carbon intensity predictions. We use the Mean Absolute Percentage Error (MAPE) to evaluate forecasting accuracy. Table 1 demonstrates a day-to-day comparison of all methods, with the most effective technique in bold. Our model demonstrates the best performance for both single-day and multi-day carbon intensity forecasting scenarios. More experimental details can be found in Appendix E.1.

## 4.2    NAS Benchmarks and Baselines

To date, there are three popular open-source NAS benchmarks, NasBench101 [90], HW-NAS-BENCH [28], and NasBench301 [93]. For our evaluation, we focus on the latter two, as the network architectures within these benchmarks cover the entirety of the search space. In contrast, the NasBench101 benchmark comprises only a limited subset of potential architectures. Evaluating using NasBench101 is challenging because we will not have access to important information, such as accuracy, during the search for the architecture not in the NasBench101 benchmark. We choose three types of baselines according to different GPU allocation strategies and NAS evaluation algorithms, including *vanilla LaMOO* [98], *one-shot LaMOO* [98] and a heuristic GPU allocation strategy [4], which is the SOTA result so far. We also define a carbon budget as the termination condition. Once the NAS exceeds this $CO_2$ budget, it will be stopped. More baseline details are in Appendix E.2.3.

### 4.2.1    Simulation using HW-NAS-Bench

Table 3: Search Results on ImageNet. The two optimization objectives we are searching for are TensorRT Latency with FP16 in NVIDIA V100 and accuracy.

| Method | Top-1 Error(%) | TensorRT Latency FP16 V100 (ms) | Search&Training Cost (GPU Hours)[†] | CO2 (lbs) | NAS Method |
|---|---|---|---|---|---|
| NASNet-A [102] | 26.0 | 2.86 | 48300 | 30679.13 | vanilla |
| PNAS [45] | 25.8 | 2.79 | 5700 | 4063.14 | vanilla |
| MnasNet [71] | 24.8 | **0.53** | 40300 | 26487.44 | vanilla |
| AlphaX [79] | 24.5 | 2.52 | 3900 | 2768.21 | vanilla |
| AmoebaNet-C [66] | 24.3 | 2.63 | 75900 | 44177.38 | vanilla |
| AutoSlim [91][‡] | 25.8 | 1.64 | 480 | 321.89 | one-shot |
| ProxylessNAS [14][‡] | 25.4 | 0.98 | 500 | 332.07 | one-shot |
| SinglePathNAS [31][‡] | 25.3 | 1.13 | 686 | 455.99 | one-shot |
| PC-DARTS [88][‡] | 24.2 | 1.50 | 411 | 278.15 | one-shot |
| FBNet-C [83][‡] | 21.5 | 1.22 | 576 | 381.58 | one-shot |
| OFA-Net [13][‡] | 20.0 | 1.16 | 1315 | 888.85 | one-shot |
| **CE-Net-G1**[‡] | 21.0 | **0.56** | 2706 | 909.86 | hybrid |
| **CE-Net-G2**[‡] | **19.4** | 0.78 | 2706 | 909.86 | hybrid |

[†] The total cost includes both the NAS search duration and the training time for the architectures identified.
[‡] The architecture is searched on ImageNet directly, otherwise it is searched on CIFAR-10 by transfer setting.

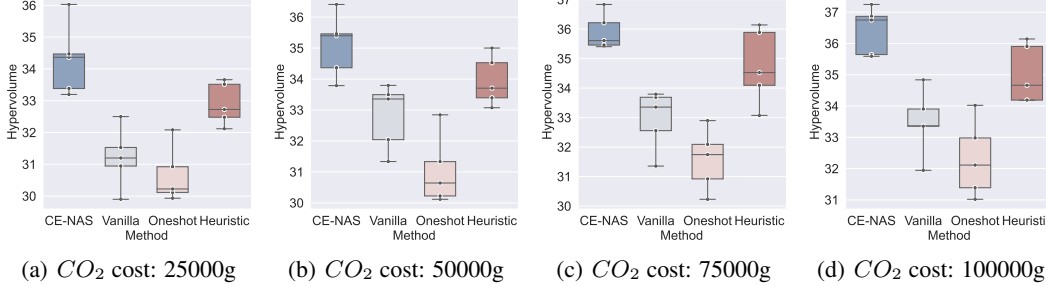

(a) $CO_2$ cost: 25000g    (b) $CO_2$ cost: 50000g    (c) $CO_2$ cost: 75000g    (d) $CO_2$ cost: 100000g

Figure 5: Search efficiency under carbon emission constraints. These results are based on Nas-Bench301 with carbon trace showing in fig. 2, and we ran each method five times.

We evaluate our experimental results using two metrics: *relative carbon emission* and *log hypervolume difference*. Note that *log hypervolume difference* measures the quality of the search result and the smaller the better. Detailed definitions of these metrics can be found in Appendix E.2.4. As depicted in Figure 3(a), as the search time progresses, vanilla LaMOO demonstrates the highest performance in terms of $HV_{\mathrm{log\_diff}}$. Vanilla LaMOO's superior performance can be attributed to its approach of *training all sampled architectures* to obtain their true accuracy, which effectively steers the search direction. However, when considering the relative carbon emission, vanilla LaMOO consumes 3.24X-7.22X carbon compared to other approaches. This is expected because vanilla LaMOO is an energy-intensive approach and is not designed to be aware of carbon emissions.

We show that CE-NAS's search efficiency is only marginally lower than that of vanilla LaMOO while having the least relative carbon emission. Note that we are plotting the $HV_{\mathrm{log\_diff}}$ in the Y-axis of Figure 3(a); the actual $HV$ values achieved by CE-NAS and Vanilla LaMOO are about 4100 and 4117, differing only by 0.034%, even though the two lines look far apart. This result also suggests that only relying on energy-efficient approaches (e.g., one-shot LaMOO in this case) is insufficient to achieve good search performance. CE-NAS only takes 15 hours to get the comparable $HV$ results with the final results of One-shot LaMOO.

Figure 4 presents a comparative analysis of CE-NAS's performance against various baselines under differing carbon budgets. It is evident that CE-NAS surpasses all baselines in search efficiency under different $CO_2$ budget constraints[5]. Below a consumption level of 10,000g $CO_2$, Notably, the search outcomes associated with CE-NAS, as depicted in Figure 4(a), and those of vanilla/one-shot LaMOO, as shown in Figure 4(d), are comparable in terms of the median log hypervolume difference. This indicates that integrating CE-NAS with the NAS strategy can achieve a carbon cost saving of up to 5X compared to the conventional vanilla/one-shot NAS methodologies.

---

[5]We did not account for the carbon cost of training and inference for the RL models and the time-series transformer, as they consume negligible $CO_2$ compared to the NAS process.

#### 4.2.2 Simulation using NasBench301

On NasBench301, we use two metrics: *relative carbon emission* and *hypervolume* ($HV$). Note that higher HV means better search results. Given the expansive nature of the NasBench301 search space, the maximal hypervolume remains undetermined. Consequently, rather than employing the hypervolume difference as a performance metric, we utilize the absolute hypervolume value to represent the efficacy of our search strategy. As depicted in Figure 3(b), our findings are consistent with those from HW-NasBench. One-shot LaMOO incurs the lowest carbon footprint but yields the least impressive performance in terms of hypervolume. Conversely, Vanilla LaMOO achieves the highest $HV$ but at a carbon cost that exceeds CE-NAS by more than 3.76 times. CE-NAS markedly surpasses the SoTA heuristic GPU allocation strategy regarding $HV$, while maintaining a comparable carbon expenditure. At equivalent levels of carbon cost, as illustrated in Figure 5, CE-NAS demonstrates significantly superior search performance compared to other baselines. Figures 5(a) and 5(d) show that architectures identified through CE-NAS not only yield better results but also do so with a carbon cost that is four times lower than that of one-shot and vanilla NAS methods. Within the NasBench301 framework, CE-NAS also substantially outperforms the heuristic GPU allocation approach for NAS.

### 4.3 Open-Domain NAS Tasks

#### 4.3.1 Searching on CIFAR-10 Image Classification Task

Our CE-NAS framework is compared with other popular NAS baselines, including vanilla and one-shot methods. Table 2 presents the SOTA results using DARTS and NASNet search spaces on CIFAR-10. The first group in the table comprises models discovered using vanilla NAS, and the second group includes those found with one-shot NAS. The $CO_2$ cost is calculated based on the duration of the search/training and the carbon intensity in the CISO region. For our CE-NAS, we select the optimal architectures from the Pareto frontier of the search results. Our CE-Net-P1 has only 1.68M parameters, significantly reducing parameter size compared to other baselines while maintaining a comparable top-1 accuracy (97.35%) with other SOTA models. Our CE-Net-P2 surpasses all baselines in top-1 accuracy (97.84%) while maintaining a similar parameter count of 3.26M. The performance discrepancy between the one-shot (second group) and vanilla NAS (first group) methods is attributed to the supernet's inaccurate accuracy prediction [31, 92]. Remarkably, our CE-NAS not only outperforms all vanilla-based NAS algorithms but also incurs a mere 38.53 lbs of $CO_2$ for the NAS search, comparable to the cost of one-shot-based NAS methods. More details related to the search space, training/search setup in More details in Appendix E.3.1.

#### 4.3.2 Searching on ImageNet Image Classification Task

Table 3 demonstrates a comparison between our CE-NAS and other state-of-the-art (SOTA) baselines. We selected our searched models, designated as CE-Net-G1 and CE-Net-G2, from the Pareto frontier, based on the dual objectives of accuracy and TensorRT latency. Our searched architectures incur a slightly higher carbon cost (909.86 lbs) compared to one-shot-based SOTA baselines but significantly outperform these baselines in terms of top-1 accuracy and TensorRT latency. More details on search space, and training/search setup are demonstrated in Appendix E.3.2.

## 5 Conclusion

In this work, we described the design of a carbon-efficient NAS framework CE-NAS by leveraging the temporal variations in carbon intensity. To search for energy-efficient architectures, CE-NAS integrates a SOTA multi-objective optimizer, LaMOO [98], with the one/few-shot and vanilla NAS algorithms. These two NAS evaluation strategies have different energy requirements, which CE-NAS leverages an RL-based agent to schedule when to use each based on average carbon intensity. CE-NAS has demonstrated very promising results across various NAS tasks. For example, on CIFAR-10, CE-NAS successfully identified an architecture that achieves 97.35% top-1 accuracy with just 1.68M parameters and emitted only 38.53 lbs of $CO_2$. These compelling results suggest the efficacy and potential of CE-NAS as an effective framework in carbon-aware NAS. Additionally, on ImageNet, CE-NAS discovered state-of-the-art models achieving a top-1 accuracy of 80.6% with a TensorRT latency of 0.78 ms using FP16 on NVIDIA V100, and a top-1 accuracy of 79.0% with a latency of 0.56 ms on the same hardware. The carbon cost for this search was only 909.86 lbs.

## 6 Acknowledgement

This work was supported in part by NSF Grants #2105564 and #2236987, a VMware grant, the Worcester Polytechnic Institute's Computer Science Department, and the National Natural Science Foundation of China under No. 62072302. This work also used Expanse at San Diego Supercomputer Center through allocation CIS230364 from the Advanced Cyberinfrastructure Coordination Ecosystem: Services & Support (ACCESS) program, which is supported by National Science Foundation grants #2138259, #2138286, #2138307, #2137603, and #2138296.

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

# A Background Knowledge

## A.1 Multi-objective NAS

Multi-objective NAS has gained popularity in the NAS community, as deep neural networks need to address not only traditional metrics like accuracy but also practical efficiency metrics. There are two main approaches within Multi-objective NAS. The first category [83, 71, 13, 23, 27] leverages linear scalarization of various metrics (e.g., $\frac{accuracy}{\#FLOPs}$) into a single metric, followed by the application of standard single-objective NAS algorithms for architecture search. The second category [98, 53, 52] approaches MONAS as a multi-objective black-box optimization problem. According to existing results [98, 52], this latter approach generally outperforms the former in terms of effectiveness and performance. For multi-objective optimization, mathematically, we optimize $M$ objectives $\boldsymbol{f}(\boldsymbol{x}) = [f_1(\boldsymbol{x}), f_2(\boldsymbol{x}), \dots, f_M(\boldsymbol{x})] \in \mathrm{r}^M$:

$$\min \quad f_1(\boldsymbol{x}), f_2(\boldsymbol{x}), ..., f_M(\boldsymbol{x}) \tag{2}$$
$$\text{s.t.} \quad \boldsymbol{x} \in \Omega,$$

where $\boldsymbol{x}$ represents the neural architecture from the search space, $f_i(\boldsymbol{x})$ denotes the function of objective $i$. Modern MOO methods aim to find the problem's entire *Pareto frontier*, the set of solutions that are not *dominated* by any other feasible solutions. Here we define *dominance* $\boldsymbol{y} \prec_{\boldsymbol{f}} \boldsymbol{x}$ as $f_i(\boldsymbol{x}) \leq f_i(\boldsymbol{y})$ for all metrics $f_i$, and there exists at least one $i$ s.t. $f_i(\boldsymbol{x}) < f_i(\boldsymbol{y}), 1 \leq i \leq M$. If the condition holds, a solution $\boldsymbol{x}$ is always better than solution $\boldsymbol{y}$.

Table 4: Comparison of energy-efficient NAS evaluation methods. *Eval. cost* refers to the cost of obtaining the evaluation results. *Init. cost* describes additional dataset preparation and the time required for training the model (e.g., supernet or predictor). *Accuracy* measures the rank correlation between the evaluation method and the actual rank. Predictor-based methods require *Extra data* as a training set to construct the prediction model.

| Method | Eval. cost | Init. cost | Accuracy | Extra data |
|---|---|---|---|---|
| Zero-shot proxy [41, 85, 50, 59, 44, 70, 16, 17] | Low | Low | Low | No |
| One-shot [46, 65, 97, 88, 13] | Low | Low | Intermediate | No |
| Predictor [45, 26, 79] | Low | High[†] | High[†] | Yes |
| Low-fidelity [102, 66, 46, 79, 36] | High | None | High[‡] | No |
| Gradient Proxy [87] | Low | Low | Intermediate | No |

[†] It depends on the size of extra data.
[‡] It depends on the extent of the fidelity.

## A.2 One (few)-shot NAS

One (few)-shot NAS utilizes a weight-sharing technique, avoiding the need to retrain sampled networks from scratch. Specifically, one-shot NAS initially trains an over-parameterized *supernet*, including all possible operations and connections within the entire search space, and employs this supernet as a performance estimator to predict an architecture's efficacy [10, 92, 97, 65]. The performance of any architecture within the search space can be approximated by leveraging the well-trained supernet. Few-shot NAS [97, 34, 86, 32, 69] further improves the accuracy of architecture evaluations conducted by supernets. It achieves this by using multiple sub-supernets, thereby reducing the co-adaptation impact among operations [10]. In this work, we employ this few-shot supernet as an efficient proxy for architecture evaluation, with more details provided in Section 3.3.

## A.3 Zero shot proxy NAS

Zero-shot NAS leverages training-free score functions/proxies to efficiently evaluate the performance of neural architectures [41, 85]. Compared to vanilla NAS and one/few-shot NAS, zero-shot NAS is cost-effective due to its training-free nature at the search stage. However, the main disadvantage of zero-shot NAS is its evaluation performance, which can be very inaccurate compared to one/few-shot NAS and vanilla NAS [41].

### A.4 Reinforcement learning (RL)

RL has been widely used for designing system strategies for tasks such as scheduling and resource allocation. For instance, RL has been deployed to develop resource allocation strategies in wireless networks [22, 89, 15, 48] and cloud platforms [19, 47, 94, 20], as well as for making bitrate decisions in adaptive video streaming systems [57, 49, 42, 82], and also generating task scheduling strategies in data centers [64, 7, 96, 8]. The key advantage of RL-based methods is their capability to learn strategies and easily adapt to system changes automatically. An RL agent begins by taking the system state as input and proceeds to generate an action, i.e. a scheduling or allocation decision, based on its learned strategies, which is typically randomized at the start. Upon execution of the action, the system provides feedback in the form of a reward, enabling the RL agent to refine its strategies. The key to applying RL lies in the design of effective state inputs, actions, and rewards according to the specific task. Our CE-NAS focuses on tailoring these elements for carbon-efficient NAS tasks, with the guiding principle of simplifying the state and action spaces to expedite efficient and rapid training.

## B  Learning Search Space Partition

We utilize LaMOO [98] to partition the search space, $\Omega$, into several sub-search spaces. This partitioning will be based on the architectures' accuracy $E(a)$ and their evaluation metrics $T(a)$ as observed in the sample set, $\mathcal{P}$. Specifically, LaMOO recursively divides the search space into promising and non-promising regions. Each partitioned region can then be mapped to a node in a search tree. Using Monte-Carlo Tree Search (MCTS), LaMOO selects the most promising sub-space (i.e., tree node) for further exploration based on their UCB values [5]. This optimal sub-space selected by MCTS is denoted as $\Omega_{best}$.

## C  Details of Reinforcement Learning Based GPU Allocation Strategy

**State Input.**  We design the RL state input at time step $t$ as $s_t = (b_t, n_t, h_t, \mathbf{c_t})$, where $b_t$ is the remaining carbon budget of the current searching task, $n_t$ is the number of already searched architectures, $h_t$ is the hypervolume of these architectures, and $\mathbf{c_t}$ is the future carbon traces in the upcoming hour (e.g., predicted by our time series transformer from §3.5.2). The general idea is that being aware of $b_t$ helps the RL algorithm plan the overall amount of carbon emission to spend for architecture evaluation, while $n_t$, $h_t$ and $\mathbf{c_t}$ help it decide the best timing to execute the evaluation that generates the highest hypervolume improvements per carbon cost.

**Network architecture.**  Our network consists of an input layer, 2 hidden layers, and an output layer. All of the layers are fully connected layers, with hidden sizes of 100, 150, 200 and 100 respectively, which are determined by performing a simple grid search. The final output of the network specifies the probability of distribution of actions.

**Action.**  We design the output of the RL agent as $K$ discrete actions, i.e. the output is a vector $\mathbf{o_t} = <o_t^1, o_t^2, ..., o_t^K>$, where $o_t^k$ denotes the probability of allocating a ratio of $\frac{(k-1)}{K}$ GPUs for architecture evaluation. We use K = 8 for our evaluation. We also implement a version of the agent with continuous action space, i.e., the network outputs the mean $\mu$ and standard $\sigma$ of a Gaussian distribution to indicate the probability distribution of the GPU allocation ratio. Our evaluation in §F.4 shows that the two versions achieve comparable performance.

**Rewards.**  As the overall goal is to maximize the cumulative reward that represents the final accuracy obtained by the search task, we design the reward in each time step as the improvement of the best accuracy of the already searched architectures: $r_t = (co_t, hi_t, nn_t)$, where $co_t$ denotes the carbon cost in this iteration, $hi_t$ represents the hypervolume increase in this iteration, and $nn_t$ means the number of new samples in this step.

**Policy gradient.**  We adopt the popular *actor-critic* algorithm to calculate the policy gradient and update the network. *Actor-critic* has been proved to achieve good performance while enabling fast training in tasks of similar scales [57, 49, 75, 67, 101]. Note that our design is not coupled with any specific DRL algorithm. Thus, replacing *actor-critic* with other algorithms is easy if needed. The key

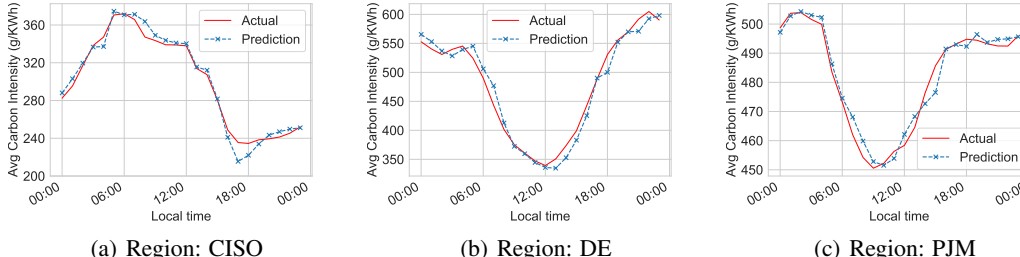

(a) Region: CISO          (b) Region: DE          (c) Region: PJM

Figure 6: Transformer-based carbon intensity predictor. Our predictor makes forecasts that perfectly match actual values over three different regions.

gradient equation of *actor-critic* is as follows:

$$\nabla E_{\pi_\theta}\left[\sum_{t=0}^{\infty}\gamma^t r_t\right] = E_{\pi_\theta}[\nabla_\theta \log \pi_\theta(s,a) A^{\pi_\theta}(s,a)] \tag{3}$$

where $\sum_{t=0}^{\infty}\gamma^t r_t$ is the cumulative rewards and $\gamma$ is a discount factor. *Actor-critic* algorithm includes two networks referred to as *actor* and *critic* respectively. In our design, they use the same network structure. The *actor* is responsible for outputting $\pi_\theta(s,a)$, which is the output probability for action $a$ with input state $s$. The policy parameter $\theta$ is the network parameter of the *actor*. $A^{\pi_\theta}(s,a)$ is the *advantage function* that indicates the performance difference between the current policy and the average performance of policies learned by *actor*. $A^{\pi_\theta}(s,a)$ is obtained with the *temporal difference* method based on the output of the *critic*. The calculation is as follows,

$$A(s_t,a_t) = r_t + \gamma V^{\pi_\theta}(s_{t+1};\theta_v) - V^{\pi_\theta}(s_t;\theta_v) \tag{4}$$

where $A(s_t,a_t)$ is an unbiased estimation of $A^{\pi_\theta}(s,a)$, $\theta_v$ is the parameter of the critic. $V^{\pi_\theta}(s_t;\theta_v)$ is the output of *critic* that estimates the cumulative reward follow actor policy $\pi_\theta$ starting from state $s_t$.

## D    Architecture Design of Time series Transformer

Building upon the foundational design of the time series Transformer, we employ LaMOO [98] as our NAS strategy to identify the optimal architectural parameters for our carbon dataset [58]. Our goal is to devise an architecture that not only achieves high forecasting accuracy but also maintains efficient training and latency times. Given that we are working with a standard encoder-decoder Transformer, our search space is straightforward. We focus on searching for: i) the number of encoder layers, ii) the number of decoder layers, iii) the embedding size, and iv) the maximum context length of the time series that the model is capable of considering. Following our NAS search process, we configured a model with 64 encoder layers, 96 decoder layers, with an embedding size of 48. Additionally, we set the context length to 8760.

## E    Experimental Details in Sec. 4

### E.1    Time-series Transformer for Carbon Forcasting

We align our training and testing data with the approach used in [55] by utilizing ElectricityMap [58], a third-party carbon information service. We select three regions (namely, CISO[6], DE[7], and PJM[8]) from ElectricityMap [58] to represent distinct carbon trace scenarios for our experiments. The dataset spans from January 1, 2020, to December 31, 2021, with an hourly resolution. We employ a train(validation)-test split of 75%–25%. We trained a single transformer model using data from three

---

[6]CISO: California Independent System Operator
[7]DE: Germany
[8]PJM: Pennsylvania-Jersey-Maryland Interconnection

different regions simultaneously. Additionally, we conducted separate training sessions for the model using distinct traces from each location. The experimental results from both approaches were similar.

In our training configuration, the context length is set to 8760, as determined by our NAS search. We utilize the AdamW optimizer, setting the weight decay to $1 \times 10^{-2}$ and selecting beta1 as 0.9 for the exponential decay rate of the first moment estimates, and beta2 as 0.95 for the second moment estimates. The model is trained over 150 epochs with an initial learning rate of $6 \times 10^{-4}$. The batch size is configured at 512 on a single NVIDIA A100 GPU.

For testing, to predict carbon intensity for the forthcoming 24 hours, we use all historical data up to the last complete 24-hour block. This block is then replaced with predictions from our model to forecast the carbon intensity for the 25th hour. Subsequent forecasts are generated by updating the most recent hour with its actual value and using the model's prediction for the next hour. We measure forecasting performance using the Mean Absolute Percentage Error (MAPE). For multi-day forecasts, we similarly update and replace values for the 24-, 48-, and 72-hour marks.

Figure 6 presents the hourly time series, averaged over a week, demonstrating both actual and predicted carbon intensities for the electricity grids in these areas. The comparison suggests that our model accurately forecasts the actual carbon trace in each region, aligning closely with the observed data.

## E.2  Experimental Details on NAS Dataset

### E.2.1  Overview of HW-NAS-Bench

HW-NAS-Bench [40] enhances the original NasBench201 dataset by including additional metrics such as inference latency, parameter size, and number of FLOPs, thereby expanding the evaluative dimensions available for NAS research. NasBench201 itself is a comprehensive open-source benchmark designed for the systematic assessment of NAS algorithms [28]. Within NasBench201, architectures are constructed by stacking cells in sequence. Specifically, each cell is composed of 4 nodes interconnected by 6 edges. Nodes represent feature maps, while edges correspond to various operations that transform one node's output into the next node's input. Operations connecting nodes include zeroization, skip-connection, 1x1 convolution, 3x3 convolution, and 3x3 average pooling. To uniquely identify each architecture, we employ a numeric encoding scheme where the five operations are represented by integers 0 through 4. Consequently, a 6-element vector represents the specific architecture.

### E.2.2  Overview of NasBench301

NasBench301 [93] serves as a surrogate benchmark for the NASNet [102] search space, which has over $10^{21}$ possible architectures. NasBench301 employs a surrogate model trained on approximately 60,000 sampled architectures to predict performance across the entire DARTS [46] search space for the CIFAR-10 image classification task. This surrogate model within NasBench301 has been shown to provide accurate regression outcomes, facilitating reliable evaluations within this vast search space. NasBench301 offers predicted accuracies from the surrogate model, while other fundamental metrics such as the number of parameters (#Params), floating-point operations per second (#FLOPs), and inference time are readily ascertainable during the evaluation phase. In this multi-objective optimization context, our goal is to simultaneously maximize inference accuracy and minimize #Params within the NasBench301 search space.

The NASNet search space includes operations such as 3x3 max pool, 3x3 convolutions, 5x5 depth-separable convolutions, and skip connections. The objective here is to identify optimal architectures for both reduction and normal cells, each comprising 4 nodes, culminating in a search space of approximately $3.5 \times 10^{21}$ architectures [93, 78]. For the CIFAR-10 task, we adopt the same encoding strategy used in the NASNet search space, representing architectures as 16-digit vectors. The first four digits denote the operations in a normal cell, digits 5-8 correspond to the concatenation patterns in the normal cell, digits 9-12 represent operations in a reduction cell, and the final four digits encode the concatenation patterns in the reduction cell.

### E.2.3 Baselines

We chose three types of baselines according to different GPU allocation strategies and NAS evaluation algorithms. During the search process, all search methods employ the state-of-the-art multi-objective optimizer, LaMOO [98]. Specifically, *one-shot LaMOO* is a method that utilizes one-shot evaluations throughout the search process. The *vanilla LaMOO* relies on actual training for architecture evaluation throughout the search.

We also provide a heuristic strategy that automatically allocates GPU resources between the sampling and evaluation processes given the carbon emissions $C_t$ at time $t$ as our baseline. This allocation is based on the energy characteristics of the processes: architecture sampling is often energy-efficient because it does not involve actual training of architectures, while architecture evaluation is often energy-consuming because it does. We assume that the maximum and minimum carbon intensities $C_{max}$ and $C_{min}$ for a future time window are known. $G_t$ denotes the total number of available GPUs. $\lambda_e$ and $\lambda_s$ represent the ratio of GPU numbers allocated to the evaluation and sampling processes at a given moment, and $\lambda_e + \lambda_s = 1$. We calculate $\lambda_s$ as $\frac{C_{cur}-C_{min}}{C_{max}-C_{min}}$, where $C_{cur}$ is the current carbon intensity. The GPU allocations for the sampling and evaluation processes are, therefore, $G_t * \lambda_s$ and $G_t * \lambda_e$. This simple heuristic allocation allows the NAS system to prioritize more energy-efficient sampling tasks during periods of higher carbon intensity, whereas, during low-carbon periods, the system will allocate more resources for energy-intensive evaluation tasks.

### E.2.4 Metrics in HW-NAS-Bench

We use two main metrics to evaluate the carbon and search efficiency of CE-NAS. First, we use *relative carbon emission* to quantify the amount of $CO_2$ each NAS method is responsible for. The relative carbon emission is calculated by summing the average carbon intensity (in gCO2/KwH) over the search process. We assume that all NAS methods use the same type of GPU whose power consumption remains the same throughout the search process. Second, we use the metric *hypervolume* (HV) to measure the "goodness" of searched samples. HV is a commonly used multi-objective optimization quality indicator [24, 25, 98] that considers all dimensions of the search objective. Given a reference point $R \in \mathrm{r}^M$, the HV of a finite approximate Pareto set $\mathcal{P}$ is the M-dimensional Lebesgue measure $\lambda_M$ of the space dominated by $\mathcal{P}$ and bounded from below by $R$. That is, $HV(\mathcal{P}, R) = \lambda_M(\cup_{i=1}^{|\mathcal{P}|}[R, y_i])$, where $[R, y_i]$ denotes the hyper-rectangle bounded by the reference point $R$ and $y_i$. A higher hypervolume denotes better multi-objective results. In this experiment, we calculate the hypervolume (HV) using the accuracy and inference energy of the searched architectures. For this dataset, we use the *log hypervolume* difference, the same as in [24, 25, 98], as our evaluation criterion for HW-NASBench, since the hypervolume difference may be minimal over the search process. Therefore, using log hypervolume allows us to visualize the sample efficiency of different search methods. We define $HV_{\text{log\_diff}} := \log(HV_{\text{max}} - HV_{\text{cur}})$ where $HV_{\text{max}}$ represents the maximum hypervolume calculated from all points in the search space, and $HV_{\text{cur}}$ denotes the hypervolume of the current samples, which are obtained by the algorithm within a specified budget. The $HV_{\text{max}}$ in this problem is 4150.7236.

For our simulation, we use the training and evaluation time costs for the architectures derived from NasBench201 [28], and inference energy costs measured on the NVIDIA Edge GPU Jetson TX2 from HW-NASBench [40]. We ran the simulation 10 times with each method.

### E.2.5 Metrics in NasBench301

Given the expansive nature of the NasBench301 search space, the maximal hypervolume remains undetermined. Consequently, rather than employing the hypervolume difference as a performance metric, we utilize the absolute hypervolume value to represent the efficacy of our search strategy.

### E.3 Experimental Details on Open-Domain NAS Tasks

### E.3.1 Details on Cifar10 Image Classification Task

***Search space overview.*** Our search space is consistent with the NASNet search space [102]. It comprises eight searchable operations: 3x3 max pooling, 3x3 average pooling, 3x3, 5x5, and 7x7 depthwise convolutions, 3x3 and 5x5 dilated convolutions, and a skip connection. The architecture consists of normal cells, which retain the size of the feature map, and reduction cells, which double

the channel size and halve the feature map resolution. Each cell integrates four nodes connected by eight different operations. The search space encompasses a total of $10^{21}$ architectures. We employ the same encoding strategy as prior studies [78].

***Search setup*** We implement low-fidelity estimation methods [102, 66, 79] to expedite the energy-consuming evaluation part in the neural architecture search process. This approach effectively accelerates the evaluation process using cost-effective approximations while largely preserving the true relative ranking of the searched architectures. Initially, we utilize early stopping for training sampled architectures for 200 epochs, in contrast to 600 epochs required for the final training phase. To further hasten training, we reduce the initial channel size from 36 to 18 and increase the batch size to 320. Additionally, the number of layers is scaled down from 24 to 16 during the search phase. To speed up the evaluation process, we use the NVIDIA Automatic Mixed Precision (AMP) library with FP16 for training during the search.

We utilize a pre-trained reinforcement learning model based on data from NasBench201 and continually tune this model during the NAS process according to task-specific data. The carbon budget for the search is set at 100 lbs.

***Training setup*** The final selected architectures are trained for 600 epochs, using a batch size of 128 and a momentum SGD optimizer with an initial learning rate of 0.025. This rate is adjusted following a cosine learning rate schedule throughout the training. Weight decay is applied for regularization.

### E.3.2   Details on ImageNet

***Search space overview.*** Our ImageNet search space is modeled after EfficientNet [72]. Specifically, it includes 5-8 stages for each architecture, with the number of stages being determined by NAS. From stages 3 to 8, the type of stage, either Fused-Inverse-Residual-Block (Fused-IRB) or Inverse-Residual-Block (IRB), is searchable. Within each stage, searchable parameters include activation type (e.g., ReLU, Swish), kernel size (e.g., 1, 3, 5, 7), number of layers ([1, 10]), expansion rate ([2, 7]), and number of channels (varies based on the stage). Additionally, our search space considers input image resolutions (e.g., 224, 288, 320, 384, 456, 528). The total search space size is approximately $10^{31}$. Our NAS search focuses on two objectives: top-1 accuracy and TensorRT latency with FP16 on an NVIDIA V100. For TensorRT latency, we fixed the workspace at 10GB for all runs and benchmarked latency using a batch size of 1 with explicit shape configuration, reporting the average latency from 1000 runs.

***Search setup*** As with CIFAR-10, we implement low-fidelity estimation methods [102, 66, 79] to accelerate the energy-consuming evaluation part in the neural architecture search process. Sampled architectures are trained for 150 epochs using early stopping, as opposed to the 450 epochs required in the final training phase. During the evaluation in the search process, we also reduce the channel size by a factor of 4. To further speed up the evaluation process, we leverage AMP with FP16 for training searched architectures.

Consistent with our CIFAR-10 approach, we employ a pre-trained reinforcement learning model based on data from NasBench201 and continually fine-tune this model during the NAS process using ImageNet task data. The carbon budget for the search is set at 1000 lbs.

***Training setup*** For each architecture in the Pareto frontier, we train it on 8 Tesla V100 GPUs with a resolution of 320x320 in our two-objective accuracy and TensorRT latency. We utilize a standard SGD optimizer with Nesterov momentum of 0.9 and set the weight decay at $3 \times 10^{-5}$. Each architecture undergoes training for a total of 450 epochs, with the initial 10 epochs serving as the warm-up period. During these warm-up epochs, we apply a constant learning rate of 0.01. The remaining epochs are trained with an initial learning rate of 0.1, using a cosine learning rate decay schedule [51], and a batch size of 1024 (i.e., 128 images per GPU). The model parameters undergo decay at a factor of 0.9997 to further enhance our models' training performance.

## F   Ablation Studies

### F.1   Effectiveness of LaMOO for NAS

We conducted ten runs of LaMOO (i.e., search space split) with a random search on the HW-NASBench dataset [40]. In addition, we performed random sampling for both the LaMOO-selected

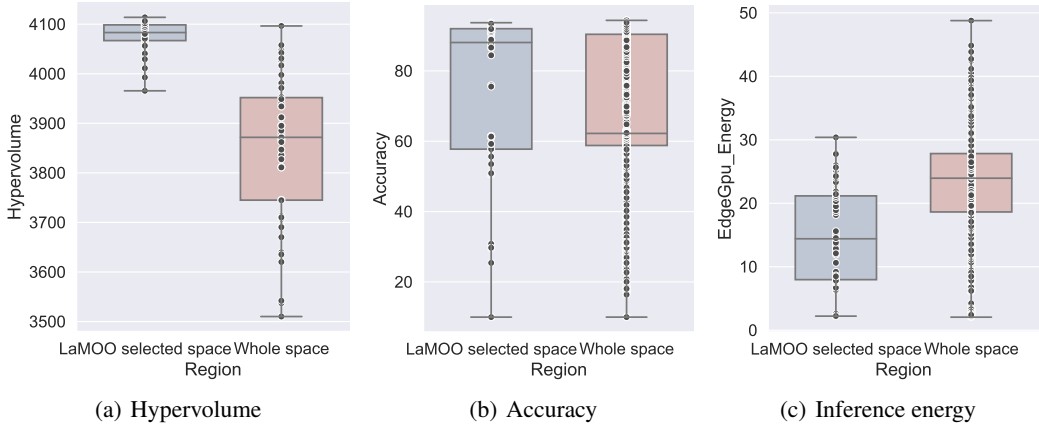

| (a) Hypervolume | (b) Accuracy | (c) Inference energy |

Figure 7: Comparisons of architecture qualities between LaMOO-selected region and the entire search space of HW-Nasbench. We ran LaMOO 10 times. For each run, we randomly sampled 50 architectures from the LaMOO-selected space and the whole search space.

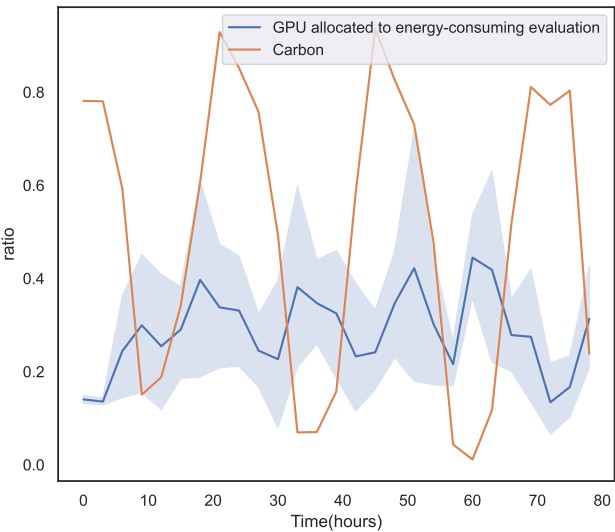

Figure 8: An overview of `CE-NAS`. This carbon trace and GPU ratio.

region and the entire search space, conducting 50 trials for each. The distribution of accuracy and edge GPU energy consumption of the architectures in both the LaMOO selected region and the entire search space can be seen in Figure 7.

Specifically, our results show that the architectures in the region selected by LaMOO have higher average accuracy and lower average edge GPU energy consumption compared to those in the entire search space. On average, the accuracy of the architectures in the LaMOO selected region is 72.12, while the accuracy in the entire search space is 68.28. The average edge GPU energy for the LaMOO selected region is 16.59 mJ, as opposed to 22.84 mJ for the entire space.

Furthermore, as illustrated in Figure 7(a), we observe that searching within the LaMOO-selected region yielded a tighter distribution, and the median hypervolume demonstrated an improvement compared to searching across the entire search space. These results suggest the efficacy of using LaMOO to partition the search space for NAS.

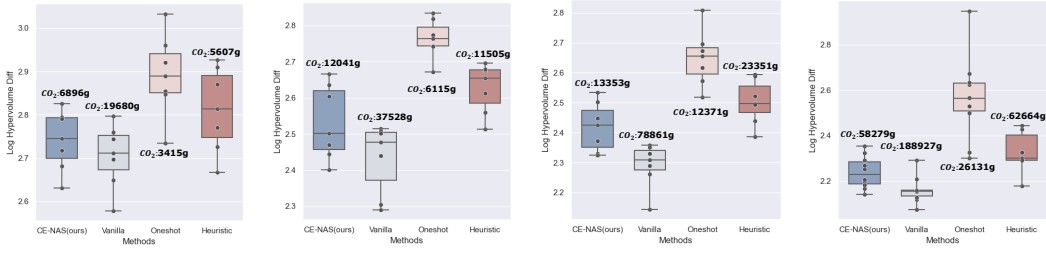

| (a) Search budget:10 hrs | (b) Search budget:20 hrs | (c) Search budget:40 hrs | (d) Search budget:80 hrs |

Figure 9: Search efficiency under time constraints. These results are based on NasBench201 with carbon trace showing in fig. 2, and we ran each method seven times.

Table 5: Search efficiency under carbon emission constraints in terms of Hypervolume. We use the transformer-based carbon predictor in sec. 3.5.2 to forecast the carbon trace and use the actual carbon trace as the baseline. We ran each method ten times and made the average.

| Carbon Constrain | CO2 Cost: 5000g | CO2 Cost: 10000g | CO2 Cost: 30000g | CO2 Cost: 50000g |
|---|---|---|---|---|
| Prediction | 3724.84 | 3774.21 | 3891.70 | 3966.70 |
| Actual | 3742.36 | 3796.93 | 3898.43 | 3971.16 |

## F.2 CE-NAS framework with Time Budget

Our framework also supports neural architecture search (NAS) within a specified time budget, in addition to the option of a carbon budget. We adapt the time budget to replace the carbon budget in the state and reward functions of reinforcement learning. Figure 9 presents the results of our Carbon-Efficient NAS (CE-NAS) under various time budgets. As illustrated, CE-NAS significantly outperforms both heuristic and one-shot methods in terms of hypervolume across different time budgets. While there is a minor decline in performance compared to vanilla NAS as measured by hypervolume, it is important to note that the carbon cost of our CE-NAS is on par with that of the heuristic method. This represents a substantial reduction compared to vanilla NAS. For instance, as depicted in Figure 9(d), vanilla NAS incurs three times the carbon cost of CE-NAS but only achieves a marginal improvement in hypervolume.

## F.3 CE-NAS framework with actual carbon trace

As detailed in Section 3.5.2, we implemented a transformer-based carbon predictor to forecast the carbon intensity for the upcoming hour, integrating this predicted value into our framework. The accuracy of this forecast is demonstrated in Fig.2. In the same section, we compare the performance of NAS searches using both the predicted carbon intensity and the actual carbon intensity. Table5 displays the results of our system operating under various carbon budget constraints with both predicted and actual carbon intensities. The data in this table indicates that, across different carbon budgets, employing the predicted carbon intensity trace yields results that are only marginally less effective than using the actual carbon trace, as measured by hypervolume.

Table 6: Search efficiency under carbon emission constraints in terms of Hypervolume. We use the continuous action space of the reinforcement learning method introduced in sec. 3.5.1. We ran each method ten times and made the average.

| RL Action Space | CO2 Cost: 5000g | CO2 Cost: 10000g | CO2 Cost: 30000g | CO2 Cost: 50000g |
|---|---|---|---|---|
| Discrete | 3724.84 | 3774.21 | 3891.70 | 3966.70 |
| Continuous | 3645.21 | 3748.83 | 3853.86 | 3920.47 |

**F.4    CE-NAS framework with continuous RL Action Space**

Recall that our RL design includes both a version with a discrete action space and another with a continuous action space. We compare their performance in Table 6 and observe that the two versions deliver comparable performance, with the continuous version performing slightly worse, experiencing a degradation within 3%. Such degradation can be attributed to that the continuous version complicates the action space and increases the learning difficulty. Despite this, we anticipate that the continuous version will outperform its discrete counterpart as the number of available GPUs grows, owing to its more fine-grained GPU allocation decision granularity that gives it the potential to generating better-optimized strategies. We leave the exploration of how GPU cluster sizes impact the relative performance of the two version for future study.

