# OpenReview forum: "CE-NAS: An End-to-End Carbon-Efficient Neural Architecture Search Framework"
_NeurIPS.cc/2024/Conference — NeurIPS 2024 poster_

### Official Review · Reviewer_QYii · 2024-07-07

**Soundness:** 4
**Presentation:** 3
**Contribution:** 4
**Rating:** 9
**Confidence:** 4

**Summary:**

The paper proposes a carbon-efficient neural architecture search (NAS), CE-NAS, to reduce the carbon emitted during the NAS process by dynamically allocate GPU resources based on the predicted future carbon intensity. CE-NAS leverages reinforcement learning for generating the probability of choosing which model of NAS (one-shot or vanilla) based on the predicted carbon intensity by a time-series transformer. One-shot model would be chosen to run in high carbon intensity periods and vanilla NAS would be chosen otherwise. Alongside emitting less carbon, CE-NAS achieves a competitive accuracy against state-of-the art models for CIFAR-10 and ImageNet datasets

**Strengths:**

1. Focusing on reducing the carbon emitted by ML tasks that has great environmental impact.
2. Designed a high-accuracy carbon intensity prediction model compared to previous models.
3. Using various baseline models in the experiments.
4. Leveraging real data for the experiments makes it more applicable to real-world scenarios.

**Weaknesses:**

1. The experiments are only conducted in California, it would be good if authors show their results for multiple locations.
2. The baseline models for comparing the proposed time-series carbon intensity predictor could use one more model proposed by Zhang et al. (Title: A GNN-based Day Ahead Carbon Intensity Forecasting Model for Cross-Border Power Grids) which is also a strong model for prediction.
3. The overhead of switching between the models has not been considered considering carbon intensity variations.

**Questions:**

1. How would the CE-NAS perform if we have a deadline for used GPU hours? e.g., we can only use 2k hours of GPU and we can't exceed that amount.
2. How robust is CE-NAS in regions that we have high carbon intensity prediction errors?

**Limitations:**

1. The authors can discuss the scalability limitations of their model when the search space becomes more complex

---

> ### Author Rebuttal · Authors · 2024-08-06
>
> We greatly appreciate the positive comments and insightful suggestions from the reviewer. Below are our answers to the reviewer's questions.
>
> ---
> **Experiments are California only**
>
> We will demonstrate more results for different locations in our future work.
>
> ---
> **Compared to the work proposed by Zhang et al [1]**
>
> Thanks for providing this useful related work. We will add this baseline in the final version of our paper.
>
> ---
> **The overhead of switching between the models has not been considered considering carbon intensity variations.**
>
> Compared to the entire NAS overhead, the carbon/energy cost of the remaining components, such as the carbon intensity predictor and reinforcement learner, is negligible.
>
> ---
> **How would the CE-NAS perform if we have a deadline for used GPU hours?**
>
> We provided an ablation study based on the time constraint in the appendix (Section F2 and Figure 9). We will move this result to the main paper if we have more space in the final version.
>
> ---
> **How robust is CE-NAS in regions that we have high carbon intensity prediction errors?**
>
> If the carbon intensity prediction is incorrect, we might allocate GPU resources inappropriately among vanilla and one-shot NAS methods. This could impact the accuracy of the searched architectures and the carbon emissions. However, the total carbon emissions should still be less than directly using vanilla NAS, and the searched architectures should still be better than directly using one-shot NAS. This is partly because of our novel use of LAMOO to refine the search space.
>
> Furthermore, we anticipate that our RL-based GPU allocation strategy will exhibit tolerance towards carbon intensity prediction errors. The adverse impact of these prediction errors will be perceived by the RL agent through its continuously updated input states, enabling it to adjust its decision-making accordingly. For instance, if the predictor underestimates carbon emission intensity, the agent may initially adopt aggressive decisions to schedule carbon-intensive tasks. This results in a rapid decrease in the remaining carbon budget, which is reflected in the input states. The RL agent perceives this change and switches to more conservative decisions in the subsequent process.
>
> ---
> **Reference**
>
> [1] Zhang, Xiaoyang, and Dan Wang. "A GNN-based Day Ahead Carbon Intensity Forecasting Model for Cross-Border Power Grids." Proceedings of the 14th ACM International Conference on Future Energy Systems. 2023.

---

> > ### Comment · Reviewer_QYii · 2024-08-12
> >
> > Thank you authors for addressing the mentioned concerns.

---

### Official Review · Reviewer_d3FP · 2024-07-11

**Soundness:** 3
**Presentation:** 3
**Contribution:** 3
**Rating:** 7
**Confidence:** 5

**Summary:**

Neural architecture search (NAS) is known to be extremely compute intensive, which also increases the corresponding energy consumption and the carbon costs due to energy production. In this work, the authors attempt to reduce the carbon cost of performing NAS. To this end, they propose a reinforcement learning (RL) agent that switches between energy intensive evaluation of architectures (standard NAS) and energy-efficient sampling (few-shot NAS), during high carbon intensive and low carbon intensity regimes, respectively. To predict the carbon intensity for efficiency scheduling of the two NAS regimes, this work uses a time series transformer which predicts the carbon intensity of energy production using historical information. Using a combination of RL agent for scheduling based on the time series transformer based carbon intensity prediction, the work shows considerable reduction in the overall carbon costs of NAS while still achieving architectures that are comparable to SOTA.

**Strengths:**

* Carbon costs of developing deep learning models is not sufficiently addressed in the literature. This work takes up a carbon efficient approach to NAS, and presents a clear motivation for looking at costs such as energy consumption and carbon emissions.

* The proposed end-to-end carbon efficient NAS that performs considerably well across benchmark and open domain tasks.

* The strategy of balancing between energy efficient sampling of architecture spaces (using few-shot NAS like approaches) and more energy intensive evaluations to explore search space better, during low and high carbon intensity times, respectively, is a strong contribution of this work. That a trade-off between these strategies is formulated using as a multi-objective optimization that aims to optimize for carbon costs is an elegant formulation of NAS than only one-shot or vanilla NAS.

* The time series predictor, although is a straightforward use of an existing transformer model, achieves good predictive performance on carbon intensity predictions.

* Experimental evaluation is sufficiently convincing. The use of real carbon intensity data from electricitymaps also provides useful insights into the real-world applications of this method.

**Weaknesses:**

* **Spatial carbon intensity variations**: This work mainly addresses the temporal variations of carbon intensity. While it can be argued that it is the first step, and also an easier mode of control, it would also be interesting to see how the scheduling can take different geographical locations into account. We can assume that large scale DL compute happens on cloud instances, and this could improve the broader usefulness of this proposed method. For instance, could the scheduler take the carbon intensity of the three regions considered in this work, and further optimize the carbon costs, as is commonly done in carbon-aware scheduling literature [4]?

* **NasBench101**: In L.256 authors state that NasBench101 does not provide accuracy. I am not sure what they mean by this. NasBench101 does provide training and validation accuracy for all the 423k architectures considered in the search space. Clarification can be useful.

* **Beyond CNNs**: This is a general critique of most NAS works, as we are exploring only CNN architectures that are by now very efficient. Why did the authors not consider other open domains? For instance, something that takes a hybrid space of CNNs and transformers [1] into account?

* **Dominance number o(a)**: If _a_ is a unique architecture (L 175), what are the _number of samples that dominate_ a given architecture _a_ (L 178)? Or is _a_ a proxy for a sub-space that is being explored? This requires some clarification. And also, on where exactly is the dominance number being used? Is it within the MOO algorithms?

#### Other comments
* L 47: Authors seem to allude that low latency is always corresponding to low energy/carbon costs. This is not always the case as has been shown in many works [5,3]
* Authors are missing some key references pertaining to carbon-aware scheduling [4] and energy/carbon efficient NAS [2,3]. While [2,3] are not exactly efficient NAS they do grapple with some of the concerns expressed in this work. These suggestions for missing reviews are only samples and not comprehensive; I encourage the authors to explore further starting from these papers.

#### References

[1] Xu, Shuying, and Hongyan Quan. "ECT-NAS: Searching efficient CNN-transformers architecture for medical image segmentation." In 2021 IEEE International Conference on Bioinformatics and Biomedicine (BIBM), pp. 1601-1604. IEEE, 2021.

[2] Dou, S., Jiang, X., Zhao, C. R., & Li, D. (2023). EA-HAS-bench: Energy-aware hyperparameter and architecture search benchmark. In The Eleventh International Conference on Learning Representations.

[3] Bakhtiarifard, P., Igel, C., & Selvan, R. (2024, April). EC-NAS: Energy consumption aware tabular benchmarks for neural architecture search. In ICASSP 2024-2024 IEEE International Conference on Acoustics, Speech and Signal Processing (ICASSP) (pp. 5660-5664). IEEE.

[4] Radovanović, A., Koningstein, R., Schneider, I., Chen, B., Duarte, A., Roy, B., ... & Cirne, W. (2022). Carbon-aware computing for datacenters. IEEE Transactions on Power Systems, 38(2), 1270-1280.

[5] Henderson, P., Hu, J., Romoff, J., Brunskill, E., Jurafsky, D., & Pineau, J. (2020). Towards the systematic reporting of the energy and carbon footprints of machine learning. Journal of Machine Learning Research, 21(248), 1-43.

**Questions:**

See weaknesses.

**Limitations:**

The authors do no discuss limitations of the work. I would encourage them to discuss the effects of the carbon intensity predictor. It could be done at two steps: Firstly, dependent on the accuracy. What if the predicted carbon intensity is inaccurate? What is the impact of scheduling the energy intensive step during carbon intensive times?

Second, at a higher level. Let us assume everyone starts doing carbon efficient NAS (or even computations in general) then there is a possible risk of rebound effect which might then go on to increase the carbon footprint paradoxically.

---

> ### Author Rebuttal · Authors · 2024-08-06
>
> We sincerely appreciate the reviewer for a thorough review and valuable comments. Below are our responses to the reviewer's concerns.
>
> ---
> **Spatial carbon intensity variations**
>
> Thank you for the suggestion. Considering the spatial variation of carbon intensity is an intriguing idea, and it is indeed feasible to further extend our scheduler to include such information in its strategy. For example, inspired by the EuroSys paper[1], the most straightforward approach is for the scheduler to first pick the data center with the lowest carbon intensity and subsequently leverage temporal variation to optimize GPU allocation.
>
> However, to further enhance scheduler performance, such as enabling dynamic switching between different data centers during the training process, we anticipate a key challenge will be managing the carbon costs associated with spatial shifts. Transferring large volumes of data over the Internet incurs significant carbon costs, and estimating these costs can be complex. The data can traverse a random number of intermediate routers across a large geographic region, making it difficult to ensure that the benefits of performing a spatial shift outweigh its overhead. We will add this discussion to the final paper and cite related work in carbon-aware scheduling.
>
> ---
> **NasBench101**
>
> We apologize for any confusion caused. What we meant is that NasBench101 does not provide accuracy for all architectures in the defined search space.  While NasBench101 does provide accuracy for 423k architectures, **the actual search space is much larger**. This means that if we use the search space defined in NasBench101, we may encounter architectures that are not included in the 423k provided by NasBench101 and therefore lack accuracy information.
>
> ---
> **Beyond CNN**
>
> We agree that exploring transformer-based architecture search space is a very important and timely research topic. As mentioned in our paper, our transformer-based carbon intensity predictor is designed by the LaMOO NAS algorithm,  running inside our CE-NAS framework (Sec. 3.5.2 and Sec. D). The NAS-designed transformer-based predictor outperformed all existing state-of-the-art baselines, as shown in Table 1 of the paper.
>
> We greatly appreciate the reviewer pointing out the good hybrid search space of CNN and transformer[32]. However, the search space described in this paper is not yet open-sourced, which makes it hard for us to investigate the hybrid search space of CNN and transformer. Despite it being a promising future direction. Moreover, for this work, due to our limited computing resources, we are unable to conduct large open-domain tasks for searching transformer-based architectures. The existing NAS benchmarks[33] are all based on CNN search spaces, so we currently lack transformer-based benchmarks to validate our CE-NAS. In future work, it will be promising to implement CE-NAS into transformer-based search spaces, such as for large language model fine-tuning, adapter (LoRA) design, and more.
>
> ---
> **Dominance number o($\alpha$)**
>
> Sorry for causing the misunderstanding. $\alpha$ refers to a unique architecture. In this paper/context, a sample means a neural architecture. To clarify, I would like to provide a toy example: we aim to search for good architectures with higher accuracy and lower inference latency. Consider we have sampled 5 architectures in the search space; one of them, referred to as $\alpha$, has a dominance number O($\alpha$). The O($\alpha$) can be calculated using the remaining four sampled architectures. Specifically, if an architecture has a higher accuracy than $\alpha$ and also a lower inference latency than $\alpha$, that means this architecture dominates $\alpha$. The number of architectures among these 5 that dominate $\alpha$ is the dominance number of $\alpha$ (O($\alpha$)). As such, an architecture with a smaller dominance number is more promising than an architecture with a higher dominance number. Now, back to CE-NAS, we use the dominance number to determine the order for sampling and evaluation. Specifically, during the sampling phase, we will sample many architectures and sort them based on their dominance number.  When CE-NAS executes the evaluation component, the evaluation order of architectures is based on the dominance number; i.e., the architecture with the lowest dominance number has the highest priority to be evaluated.
>
> ---
> **L47**
>
> Our intention was to convey that we are not only focusing on performance metrics like accuracy. We can consider more metrics such as inference latency, #PARAMs, #FLOPs, etc. We will rewrite this part to clarify our meaning and remove any misleading information.
>
> ---
> **More related works**
>
> Thanks for providing useful resources. We carefully read the suggested papers and further explore starting from these sources. We have organized the papers into three categories as detailed below:
>
> + Category one[2,3,4,5,6]: This category focuses on measuring carbon emissions and the impact of carbon intensity on AI training or NAS tasks but does not include any designs to reduce carbon emissions.
>
> + Category two[7,8,9]: This category proposes energy-aware scheduling for NAS tasks but omits the effect of carbon intensity variations on carbon costs.
>
> + Category three[10,11,12,13,14]: This category considers the effect of either temporal or spatial carbon intensity variations on the carbon emissions of data center tasks, relying on inter-job execution order scheduling and location selection to reduce total emissions. However, the scheduling techniques proposed by these works to exploit temporal carbon intensity variation do not apply when optimizing for a single NAS task.
>
> We will include them in the final paper.
>
> ---
> **Limitations**
>
> Thank you for your suggestion. We will add the discussion of the carbon intensity predictor in the final paper.
>
> ---
> **Reference**
>
> Due to the characters limit, the references can be found in **Common Rebuttal for All Reviewers**.

---

> > ### Comment · Reviewer_d3FP · 2024-08-12
> > **Response to author rebuttal**
> >
> > The authors have addressed most of my concerns in their rebuttal with appropriate clarifications. I urge them include these clarifications in the final version of the paper.
> >
> > I will raise my score to Accept.

---

### Official Review · Reviewer_g77N · 2024-07-11

**Soundness:** 4
**Presentation:** 3
**Contribution:** 4
**Rating:** 8
**Confidence:** 4

**Summary:**

This paper introduces a NAS framework, named CE-NAS, which focuses on reducing the carbon footprint of architecture search. To do this, CE-NAS decides when to use which NAS evaluation method depending the current carbon intensity, and how to allocate available GPU resources. Their carbon forecasting model also outperforms previous models for predicting carbon intensity. In experiments they show that CE-NAS outperforms other methods, like vanilla NAS and one-shot NAS, given a fixed carbon budget.

Sampling and evaluation are decoupled in the framework. Sampling is done using a BO algorithm which generates new architectures. These are then added to a queue of architectures that will be put on the available GPU resources for full training. A proxy evaluation is also done before this using the supernet constructed from the most promising region of the search space, according to a multi-objective optimiser. This approximate performance is used by the BO sampling algorithm.

During periods of high carbon emissions, the sampling is the priority, and in periods of low carbon emissions, the full training is prioritised. In the latter, it also updates the promising search space region used to construct the supernet. Every hour, the priority is reassigned and GPU resources are reallocated.

**Strengths:**

**Originality**
Constructing a framework which puts carbon efficiency at its core is an original idea that goes further than previous work. The paper also motivates this idea very well.

**Quality**
The proposed solution is effective at providing models on the Pareto frontier of the optimised objectives. It consistently returns architectures that have either better performance or lower cost/parameter count and these choices are produced in the same search, allowing the user to choose between them at test time. Additionally, the carbon cost of the framework lies between vanilla and one-shot methods while being able to outperform both in terms of accuracy.

One of my concerns was that this paper presents a complex framework with many moving parts, including several trained networks. Fortunately there are good ablations of these components in the appendix, accounting for their contribution to the whole.

**Clarity**
The paper is clear and well presented overall. There are some minor issues on descriptive clarity which I have listed in the weaknesses section below.

**Significance**
I think the contribution is significant, but have some minor concerns whether the scope of this paper fits best within the NeurIPS conference. Would this paper that focuses on a complex and practical framework perhaps be better suited to a venue that focused on the practical aspects on running NAS, such as the AutoML conference? On the other hand I think the paper tackles important questions for the NAS community which should be highlighted at a venue that gives it more impact. This concern does not overall impact my rating but I would like to hear the authors thoughts on it.

**Weaknesses:**

My main concern is that the paper does not make it clear enough how a practitioner would use the framework themselves. The critical point is that real-time carbon emission data needs to be available locally and there is no discussion of how realistic this is or any suggestion for where to find it. Without this aspect of carbon efficiency, there is still a decent contribution of a framework that seems to outperform other methods for multi-objective NAS but I would like more discussion on how I could realistically use the full framework myself.

Minor comments:
* Figure 1 needs a more descriptive caption that clarifies the diagram and how CE-NAS works on a high level. The diagram is complex and hard to parse at the moment.
* The CIFAR10 performance differs between the abstract (97.35%) and the introduction (97.84%). In the end of the introduction, in the list of contributions, it is again 97.35%. This seems to be the two different models from the Pareto frontier. I suggest either explaining that you find both or be consistent with which you highlight in the text.
* Is there a citation for the claim on lines 97-99? *A recent NAS work on transformer revealed that their comprehensive architecture search used 979 million training steps, totaling 274,120 GPU hours and resulting in 626,155 pounds of CO2 emissions.*
* When reading, I was confused by both the relative carbon emission and hypervolume metrics. I think there should be a short explanation in the main paper along with the pointer to the full description in the appendix. This pointer is currently in section 4.2.1 but hypervolume is first mentioned in 3.5.1 which causes confusion for a while.
* The caption for Figure 2 states that CE-NAS has the lowest relative carbon emission but this seems to be wrong. One-shot LaMOO has a lower CO_2 number in (a) and both One-shot LaMOO and Heuristic have lower numbers in (b).
* On line 263: *“To date, there are three popular open-source NAS datasets, NasBench101 [77], HW-NAS-BENCH [25], and NasBench301 [80]”* There are other popular ones, e.g. DARTS, and I think a more appropriate term here is NAS benchmarks instead of datasets.
* Table 2, row CE-Net-P1 says hybird instead of hybrid.
* In Table 3, MnasNet achieves a TensorRT Latency of 0.53, which is lower than CE-Net-G1 and should be the bolded number (unless there was a typo).
* Figure 9 of the appendix needs a caption that better explains it.

**Questions:**

How easy is it in practice to access live information about carbon efficiency? This is key to the method and reducing carbon emissions from the NAS process, but as a practitioner I would not know how to get this information. If there is a straightforward way to get this data from, e.g. ElectricityMap, I would like to see that described in the paper or pointed to in the appendix. Additionally, it would be good to have a discussion on the availability of carbon emissions data across different countries/regions as I imagine it will be unavailable in many places.

**Limitations:**

There is no clear section that discusses limitations, and I think this is where my suggestion on discussing the availability of carbon emissions data could be put.

---

> ### Author Rebuttal · Authors · 2024-08-06
>
> We sincerely thank the reviewer for your insightful comments, please check our answers below.
>
> ---
> **How easy is it in practice to access live information about carbon efficiency?**
>
> There are third-party providers, such as ElectricityMap and WattTime, that allow users to query carbon intensity through APIs [APIs1] [APIs2]. As sustainability becomes an increasingly important issue in computing, we anticipate that more resources for obtaining carbon information will become available. We will include additional details about the accessibility of carbon information in the final paper.
>
> [APIs1] https://static.electricitymaps.com/api/docs/index.html
>
> [APIs2] https://docs.watttime.org/#tag/GET-Regions-and-Maps/operation/get_reg_loc_v3_region_from_loc_get
>
> ---
> **Discussion on the availability of carbon emissions data across different countries/regions**
>
> You are correct that carbon data signals are only available in certain parts of the world. For example, there is a lack of such information for Africa and large parts of Asia [CarbonAvailability]. However, we believe there will be a growing range of grid regions as commercial services flourish.
>
> Currently, in our work, CE-NAS leverages the temporal variation of carbon signals to strategically allocate GPU resources for NAS methods. This means that CE-NAS can be used on GPU clusters that operate in regions with available carbon emission data. Users from regions that currently do not have these carbon signals can still contribute to sustainability goals by launching CE-NAS on remote GPU clusters.
>
> [CarbonAvailability] https://watttime.org/docs-dev/coverage-map/
>
> ---
> **Make Figure 1 better**
>
> We will simplify this figure and write a descriptive caption based on your suggestion.
>
> ---
> **Inconsistency in the abstract and the introduction**
>
> In abstract, we want to highlight we searched small model still has high accuracy. In introduction, we want to emphasis that we can find state-of-the-art models in terms of accuracy. Sorry for making this confusing inconsistency in the different context of the paper. We will make them consistent in our paper.
>
> ---
> **Citation for the claim on lines 97-99**
>
> It shares the same citation ([60] in the paper) with the last sentence. We will add this citation to the paper in this sentence.
>
> ---
> **Definition of carbon emission and hypervolume metrics**
>
> Thanks for catching this. We will move he definition of relative carbon emission and hypervolume metrics to a more appropriate place in the paper.
>
> ---
> **Inaccurate caption statement in Figure 2**
>
> Thanks for catching this. We will fix it in our final version.
>
> ---
> **Appropriate term for NasBench**
>
> We will rephrase the dataset to benchmarks in our final version.
>
> ---
> **MnasNet should be bolded in Table 3**
>
> We will bold it as the lowest TensorRT Latency in the table.
>
> ---
>
> **Caption of Figure 9**
>
> Figure 9 describes the search efficiency among different NAS methods under time constraints. The total carbon cost of each method is shown above each method. We will also add more details in the figure caption.

---

> > ### Comment · Reviewer_g77N · 2024-08-12
> > **Response to authors**
> >
> > Thanks to the authors for their detailed response. After reading this and the other reviews and responses, I think the paper will be improved. On the assumption that these changes will be made for the final version, have decided to upgrade my score.

---

### Official Review · Reviewer_CjFX · 2024-07-14

**Soundness:** 3
**Presentation:** 3
**Contribution:** 2
**Rating:** 5
**Confidence:** 5

**Summary:**

The paper introduces CE-NAS, a novel framework for neural architecture search that prioritizes carbon efficiency in the model design process. It addresses the high carbon cost associated with NAS by dynamically adjusting GPU resources based on predicted carbon intensity and search results. The framework integrates a reinforcement-learning agent, a time-series transformer for carbon intensity prediction, and a multi-objective optimizer to balance energy-efficient sampling with energy-intensive evaluation tasks. Experiments demonstrate CE-NAS's ability to reduce carbon emissions significantly while achieving good results on various NAS datasets and tasks.

**Strengths:**

* CE-NAS presents a unique and timely approach to NAS that considers carbon efficiency, aligning with broader environmental sustainability goals.
* The integration of a reinforcement-learning agent, time-series transformer, and multi-objective optimizer provides a comprehensive solution for carbon-aware NAS.
* The framework's effectiveness on various datasets suggests that it could be widely applicable to different NAS scenarios.

**Weaknesses:**

* Experimental comparisons are focused on traditional and one-shot NAS methods. However, zero-shot NAS methods are not mentioned in the paper. Some recent zero-shot NAS methods have achieved better results and higher efficiency.
* The integration of multiple components introduces complexity in implementation.

**Questions:**

Please describe the advantages of CE-NAS when compared to zero-shot NAS methods.

---

> ### Author Rebuttal · Authors · 2024-08-04
>
> We sincerely thank the reviewer for your insightful comments and suggestions, please check our answers below.
>
> ---
>
> **Comparison between CE-NAS and zero-shot NAS**
>
> We first thank the reviewer for pointing out the missing related work on zero-shot NAS in our paper.  We will add the following discussion, comparison, and potential integration with zero-shot NAS in our final version.
>
> First, Zero-shot NAS leverages training-free score functions/proxies to efficiently evaluate the performance of neural architectures [15][16]. Compared to vanilla NAS and one/few-shot NAS, zero-shot NAS is cost-effective due to its training-free nature at the search stage. However, the main disadvantage of zero-shot NAS is its evaluation performance, which can be very inaccurate compared to one/few-shot NAS and vanilla NAS. According to [15], mainstream zero-shot NAS methods, such as gradient norm [17], SNIP [18], Synflow [19], GraSP [20], GradSign [21], Fisher [22], Jacobian_cov [23][24], Zen_score [25][26], and NTK [27][28] did not achieve high rank correlations between architectures’ predicted and true accuracy. For example, in the NasBench201 search space for CIFAR10, the highest Kendall’s tau value (a ranking correlation metric ranging from -1 to 1, where 1 indicates identical ranking and -1 indicates completely reversed ranking) from these zero-shot methods is below 0.6 [21], with most values ranging from 0 to 0.5. In contrast, one/few-shot NAS [29][30] can achieve Kendall’s tau value of 0.75. Importantly, for the top 5% of architectures ordered by true accuracy, most zero-shot methods [17,18,19,20,22,23,24,27,28] have Kendall’s tau value around or below 0, equivalent to random search. Zen-score [25, 26] and GradSign [21] perform slightly better than others but still below 0.3. Based on our experiments, few-shot NAS [29] achieves Kendall’s tau value of 0.56 for the top 5% accurate architectures, significantly higher than zero-shot NAS.
>
> Second, Our CE-NAS integrates few-shot NAS and vanilla NAS, leveraging vanilla NAS in a low-carbon intensity period to provide accurate performance feedback, effectively guiding the search algorithms and identifying promising regions in the search space. Our CE-NAS effectively guides the exploration of optimal neural architectures with minimal carbon emissions. Zero-shot NAS, due to its high randomness and inaccurate performance proxies, struggles to find satisfactory neural architectures. Additional evidence from [15] shows that the top-1 accuracy of the best zero-shot NAS method [27, 28] on the ProxylessNAS search space is 73.63%, only slightly better than random search sorted by the highest FLOPs (73.08%). In comparison, one-shot-based ProxylessNAS[31] achieves 75.1% top-1 accuracy, and few-shot ProxylessNAS further improves this to 75.91%[29]. Our CE-NAS achieves a top-1 accuracy higher than 80% on ImageNet.
>
> In summary, we acknowledge that zero-shot NAS can be very cost-effective and energy/carbon-efficient. However, relying solely on training-free proxies to evaluate neural architectures can be extremely inaccurate, compared to one/few-shot NAS. Nevertheless, we note that CE-NAS is complementary to zero-shot NAS. Specifically, CE-NAS can integrate zero-shot NAS methods for quick evaluations, replacing one/few-shot NAS. The vanilla NAS component is a valuable complement that helps zero-shot NAS search in promising regions, thereby improving the search effectiveness of zero-shot NAS.
>
> ---
> **The integration of multiple components introduces complexity in implementation.**
>
> We appreciate reviewer g77N’s attention to this matter and would like to highlight that we have conducted many ablation studies of CE-NAS components in the appendix (Sec. F). These ablation studies provide strong evidence that these components work well together and are indispensable to the framework. Additionally, compared to the entire NAS overhead, the carbon/energy cost of the remaining components, such as the carbon intensity predictor and reinforcement learner, is negligible. Lastly, we have included the source code as supplementary material,  and we invite the reviewer to inspect our research prototype, which we will open-source upon acceptance.
>
> ---
> **Reference**
>
> Due to a limit of 6000 characters, the reference can be found in **Common Rebuttal for All Reviewers**.

---

> > ### Comment · Reviewer_CjFX · 2024-08-13
> > **Thanks for the efforts.**
> >
> > Thanks for the response. I acknowledge that zero-shot NAS has its limitations and agree that integrating it into the CE-NAS framework is a promising direction. The work's novel combination with carbon efficiency indeed offers insights, and as such, I will increase my score.
> >
> > I suggest the authors to include a discussion on zero-shot NAS in the related work section to ensure the completeness of the paper. Additionally, the provided code in the Supp. is currently inaccessible, requiring an application for access. I kindly urge the authors to make the code publicly available.

---

> ### Author Response · Authors · 2024-08-13
>
> Thank you for your kind feedback. We apologize for the code being inaccessible earlier. We have now made it public, and we will organize and release the code on GitHub once the paper is accepted.

---

### Author Rebuttal · Authors · 2024-08-04

**Reference**

[1] On the Limitations of Carbon-Aware Temporal and Spatial Workload Shifting in the Cloud, Sukprasert et al. EuroSys 2024.

[2] Anthony L F W, Kanding B, Selvan R. "Carbontracker: Tracking and predicting the carbon footprint of training deep learning models". arXiv preprint arXiv:2007.03051, 2020.

[3] Selvan R, Bhagwat N, Wolff Anthony L F, et al. "Carbon footprint of selecting and training deep learning models for medical image analysis". International Conference on Medical Image Computing and Computer-Assisted Intervention, 2022.

[4] Naidu R, Diddee H, Mulay A, et al. "Towards quantifying the carbon emissions of differentially private machine learning". arXiv preprint arXiv:2107.06946, 2021.

[5] Lacoste A, Luccioni A, Schmidt V, et al. "Quantifying the carbon emissions of machine learning". arXiv preprint arXiv:1910.09700, 2019.

[6] Dodge J, Prewitt T, Tachet des Combes R, et al. "Measuring the carbon intensity of ai in cloud instances". Proceedings of the 2022 ACM conference on fairness, accountability, and transparency, 2022.

[7] Frey N C, Zhao D, Axelrod S, et al. "Energy-aware neural architecture selection and hyperparameter optimization". IPDPSW, 2022.

[8] Dou S, Jiang X, Zhao C R, et al. "EA-HAS-bench: Energy-aware hyperparameter and architecture search benchmark". The Eleventh ICLR. 2023.

[9] Bakhtiarifard P, Igel C, Selvan R. "EC-NAS: Energy consumption aware tabular benchmarks for neural architecture search". ICASSP, 2024.

[10] Radovanović A, Koningstein R, Schneider I, et al. "Carbon-aware computing for datacenters". IEEE Transactions on Power Systems, 2022

[11] Souza A, Jasoria S, Chakrabarty B, et al. "CASPER: Carbon-Aware Scheduling and Provisioning for Distributed Web Services". Proceedings of the 14th International Green and Sustainable Computing Conference, 2023.

[12] Wiesner P, Behnke I, Scheinert D, et al. "Let's wait awhile: How temporal workload shifting can reduce carbon emissions in the cloud." Proceedings of the 22nd International Middleware Conference. 2021

[13] Bahreini T, Tantawi A, Youssef A. "A Carbon-aware Workload Dispatcher in Cloud Computing Systems." 16th CLOUD, 2023.

[14] Tripathi S, Kumar P, Gupta P, et al. "Workload Shifting Based on Low Carbon Intensity Periods: A Framework for Reducing Carbon Emissions in Cloud Computing." BigData, 2023.

[15] Li, Guihong, et al. "Zero-Shot Neural Architecture Search: Challenges, Solutions, and Opportunities." IEEE Transactions on Pattern Analysis and Machine Intelligence (2024).

[16] Meng-Ting Wu, Chun-Wei Tsai, Training-free neural architecture search: A review, ICT Express, Volume 10, Issue 1, 2024, Pages 213-231, ISSN 2405-9595,

[17] M. S. Abdelfattah, A. Mehrotra, Ł. Dudziak, and N. D. Lane, “Zerocost proxies for lightweight nas,” in International Conference on Learning Representations, 2021.

[18] N. Lee, T. Ajanthan, and P. Torr, “SNIP: SINGLE-SHOT NETWORK PRUNING BASED ON CONNECTION SENSITIVITY,” in International Conference on Learning Representations, 2019.

[19] H. Tanaka, D. Kunin, D. L. Yamins, and S. Ganguli, “Pruning neural networks without any data by iteratively conserving synaptic flow,” in Advances in Neural Information Processing Systems, H. Larochelle, M. Ranzato, R. Hadsell, M. Balcan, and H. Lin, Eds., vol. 33. Curran Associates, Inc., 2020, pp. 6377–6389.

[20] C. Wang, G. Zhang, and R. B. Grosse, “Picking winning tickets before training by preserving gradient flow,” in International Conference on Learning Representations. OpenReview.net.

[21] Z. Zhang and Z. Jia, “Gradsign: Model performance inference with theoretical insights,” in International Conference on Learning Representations, 2022.

[22] L. Theis, I. Korshunova, A. Tejani, and F. Huszar, “Faster gaze ” prediction with dense networks and fisher pruning,” CoRR, vol. abs/1801.05787, 2018.

[23] V. Lopes, S. Alirezazadeh, and L. A. Alexandre, “Epe-nas: Efficient performance estimation without training for neural architecture search,” in International Conference on Artificial Neural Networks. Springer, 2021, pp. 552–563.

[24] J. Mellor, J. Turner, A. Storkey, and E. J. Crowley, “Neural architecture search without training,” in International Conference on Machine Learning. PMLR, 2021, pp. 7588–7598.

[25] M. Lin, P. Wang, Z. Sun, H. Chen, X. Sun, Q. Qian, H. Li, and R. Jin, “Zen-nas: A zero-shot nas for high-performance image recognition,” in Proceedings of the IEEE/CVF International Conference on Computer Vision, 2021, pp. 347–356.

[26] Z. Sun, M. Lin, X. Sun, Z. Tan, H. Li, and R. Jin, “MAE-DET: revisiting maximum entropy principle in zero-shot NAS for efficient object detection,” in International Conference on Machine Learning, ICML 2022, 17-23, Proceedings of Machine Learning Research, vol. 162. PMLR, 2022.

[27] W. Chen, X. Gong, and Z. Wang, “Neural architecture search on imagenet in four gpu hours: A theoretically inspired perspective,” in International Conference on Learning Representations, 2021.

[28] W. Chen, W. Huang, X. Du, X. Song, Z. Wang, and D. Zhou, “Autoscaling vision transformers without training,” in International Conference on Learning Representations, 2022.

[29] Zhao, Yiyang, et al. "Few-shot neural architecture search." International Conference on Machine Learning. PMLR, 2021.

[30] Su, Xiu, et al. "K-shot nas: Learnable weight-sharing for nas with k-shot supernets." International Conference on Machine Learning. PMLR, 2021.

[31] Cai, H., Zhu, L., and Han, S. ProxylessNAS: Direct neural architecture search on target task and hardware. In International Conference on Learning Representations, 2019.

[32]  Xu, Shuying, and Hongyan Quan. "ECT-NAS: Searching efficient CNN-transformers architecture for medical image segmentation." In 2021 IEEE International Conference on Bioinformatics and Biomedicine (BIBM), pp. 1601-1604. IEEE, 2021.

[33] Chitty-Venkata, Krishna Teja, et al. "Neural architecture search benchmarks: Insights and survey." IEEE Access 11 (2023): 25217-25236.

---

### Decision · Program_Chairs · 2024-09-25

**Decision:**

Accept (poster)

**Comment:**

This paper introduces CE-NAS, a NAS (Neural Architecture Search) framework designed to minimize the carbon footprint of architecture searches. CE-NAS optimizes the use of GPU resources and NAS evaluation methods based on real-time carbon intensity forecasts, which their model predicts more accurately than existing methods. All the reviewers are satisfied with the authors' response. The authors should consider including a discussion on zero-shot NAS in the related work section to ensure the completeness of the paper.